# Effects of germline and somatic events in candidate BRCA-like genes on breast-tumor signatures

**Weston R. Bodily**[1], **Brian H. Shirts**[2], **Tom Walsh**[3,4], **Suleyman Gulsuner**[3,4], **Mary-Claire King**[3,4], **Alyssa Parker**[1], **Moom Roosan**[5], **Stephen R. Piccolo**[1] *

**1** Department of Biology, Brigham Young University, Provo, UT, United States of America, **2** Department of Laboratory Medicine, University of Washington, Seattle, Washington, United States of America, **3** Division of Medical Genetics, Department of Medicine, University of Washington, Seattle, Washington, United States of America, **4** Department of Genome Sciences, University of Washington, Seattle, Washington, United States of America, **5** Pharmacy Practice Department, Chapman University School of Pharmacy, Irvine, CA, United States of America

* stephen_piccolo@byu.edu

**Data Availability Statement:** The datasets generated and analyzed during the current study are available in the Open Science Framework repository (https://osf.io/9jhr2). These data were generated by third parties; we did not play a role in

## Abstract

Mutations in *BRCA1* and *BRCA2* cause deficiencies in homologous recombination repair (HR), resulting in repair of DNA double-strand breaks by the alternative non-homologous end-joining pathway, which is more error prone. HR deficiency of breast tumors is important because it is associated with better responses to platinum salt therapies and PARP inhibitors. Among other consequences of HR deficiency are characteristic somatic-mutation signatures and gene-expression patterns. The term "BRCA-like" (or "BRCAness") describes tumors that harbor an HR defect but have no detectable germline mutation in *BRCA1* or *BRCA2*. A better understanding of the genes and molecular events associated with tumors being BRCA-like could provide mechanistic insights and guide development of targeted treatments. Using data from The Cancer Genome Atlas (TCGA) for 1101 breast-cancer patients, we identified individuals with a germline mutation, somatic mutation, homozygous deletion, and/or hypermethylation event in *BRCA1*, *BRCA2*, and 59 other cancer-predisposition genes. Based on the assumption that BRCA-like events would have similar downstream effects on tumor biology as *BRCA1/BRCA2* germline mutations, we quantified these effects based on somatic-mutation signatures and gene-expression profiles. We reduced the dimensionality of the somatic-mutation signatures and expression data and used a statistical resampling approach to quantify similarities among patients who had a *BRCA1/BRCA2* germline mutation, another type of aberration in *BRCA1* or *BRCA2*, or any type of aberration in one of the other genes. Somatic-mutation signatures of tumors having a non-germline aberration in *BRCA1/BRCA2* (n = 80) were generally similar to each other and to tumors from *BRCA1/BRCA2* germline carriers (n = 44). Additionally, somatic-mutation signatures of tumors with germline or somatic events in *ATR* (n = 16) and *BARD1* (n = 8) showed high similarity to tumors from *BRCA1/BRCA2* carriers. Other genes (*CDKN2A*, *CTNNA1*, *PALB2*, *PALLD*, *PRSS1*, *SDHC*) also showed high similarity but only for a small number of events or for a single event type. Tumors with germline mutations or hypermethylation of *BRCA1* had relatively similar gene-expression profiles and overlapped considerably

generating the data but rather used it for secondary research. Furthermore, we did not have any special access privileges to the data; we obtained the data in the ways stated in this article. We are not permitted to share the germline-mutation data, but researchers can request access via the TCGA data access committee (https://www.cancer.gov/about-nci/organization/ccg/research/structural-genomics/tcga/contact).

**Funding:** Funding for this study was provided through Brigham Young University Graduate Studies and the Simmons Center for Cancer Research. In addition, we acknowledge grant support from NIH 1R35CA197458, Komen Foundation SAC110020, and Breast Cancer Research Foundation BCRF18-088.

**Competing interests:** TW consults for Color Genomics. Otherwise, the authors declare that they have no competing interests. This does not alter our adherence to PLOS ONE policies on sharing data and materials.

with the Basal-like subtype; but the transcriptional effects of the other events lacked consistency. Our findings confirm previously known relationships between molecular signatures and germline or somatic events in *BRCA1/BRCA2*. Our methodology represents an objective way to identify genes that have similar downstream effects on molecular signatures when mutated, deleted, or hypermethylated.

## Introduction

Approximately 1–5% of breast-cancer patients carry a pathogenic germline variant in either *BRCA1* or *BRCA2* [1–5]. These genes play important roles in homologous recombination repair (HR) of double-stranded breaks and stalled or damaged replication forks [6, 7]. When the *BRCA1* or *BRCA2* gene products are unable to perform HR, cells may resort to non-homologous end-joining, a less effective means of repairing double-stranded breaks, potentially leading to an increased rate of DNA mutations [8–11]. Patients who carry biallelic loss of *BRCA1* and *BRCA2* due to germline variants and/or somatic events often respond well to poly ADP ribose polymerase (PARP) inhibitors and platinum-salt therapies, which increase the rate of DNA damage, typically causing the cells to enter programmed cell death [12–17].

The downstream effects of BRCA mutations are distinctive. For example, BRCA-mutant tumors exhibit an abundance of C-to-T transitions across the genome, potentially reflecting tumor cells' impaired ability to repair specific types of DNA damage [18]. In large-scale sequencing projects, such mutational patterns (termed "somatic-mutation signatures") have been observed in association with other types of molecular events, as well as environmental and endogenous exposures, across many cancer types [19, 20]. Among these signatures, the so-called "Signature 3" has been associated with BRCA mutations and HR [19, 21].

Other downstream effects of BRCA mutations include characteristic transcriptional responses. For example, it has been shown that the "Basal" gene-expression subtype is enriched for tumors with *BRCA1* mutations [22–25], that *BRCA1* mutations are commonly found in triple-negative breast tumors [26, 27], and that gene-expression profiles may predict PARP inhibitor responses [28, 29]. These patterns are consistently observable, even in the presence of hundreds of other mutations [25, 30].

In 2004, Turner, et al. coined the term "BRCAness" to describe patients who do not have a pathogenic germline variant in *BRCA1* or *BRCA2* but who have developed a tumor with an impaired ability to perform HR [31]. Here we use the alternative term "BRCA-like." This category may be useful for clinical management of patients and especially for predicting treatment responses [31, 32]. Recent estimates suggest that the proportion of breast-cancer patients who fall into this category may be as high as 20% [33]. Davies, et al. demonstrated an ability to categorize patients into this category with high accuracy based on high-level mutational patterns [33]. Polak, et al. confirmed that somatic mutations, large deletions, and DNA hypermethylation of *BRCA1* and *BRCA2* are reliable indicators of being BRCA-like [21, 34–36]. They also showed a relationship between being BRCA-like and germline mutations in *PALB2* and hypermethylation of *RAD51C* [21]. However, a considerable portion of breast tumors with HR deficiency lack a known driver. Furthermore, less is known about whether the downstream effects of germline variants, somatic variants, large deletions, and hypermethylation are similar to each other or whether these effects are similar for different genes.

An underlying assumption of the BRCA-like concept is that the effects of HR deficiency are similar across tumors, regardless of the genes that drive those deficiencies and despite considerable variation in genetic backgrounds, environmental factors, and the presence of other

driver mutations. Based on this assumption—and in a quest to identify additional genes that contribute to being BRCA-like—we performed a systematic evaluation of multiomic and clinical data from 1101 patients in The Cancer Genome Atlas (TCGA) [25]. In performing these evaluations, we characterized each tumor using two types of molecular signature: 1) weights that represent the tumor's somatic-mutation profile and 2) the tumor's mRNA expression profile. To evaluate similarities among tumors based on these molecular profiles, we used a statistical-resampling approach designed to quantify similarities among patient subgroups, even when those subgroups are small, thus helping to account for rare events. We use "aberration" as a general term to describe germline mutations, somatic mutations, copy-number deletions, and hypermethylation events.

## Methods

### Data preparation and filtering

We obtained breast-cancer data from TCGA for 1101 patients in total. To determine germline-mutation status, we downloaded raw sequencing data from CGHub [37] for normal (blood) samples. We limited our analysis to whole-exome sequencing samples that had been sequenced using Illumina Genome Analyzer or HiSeq equipment. Because the sequencing data were stored in BAM format, we used Picard Tools (SamToFastq module, version 1.131, http://broadinstitute.github.io/picard) to convert the files to FASTQ format. We used the Burrows-Wheeler Alignment (BWA) tool (version 0.7.12) [38] to align the sequencing reads to version 19 of the GENCODE reference genome (hg19 compatible) [39]. We used sambamba (version 0.5.4) [40] to sort, index, mark duplicates, and flag statistics for the aligned BAM files. In cases where multiple BAM files were available for a single patient, we used bamUtil (version 1.0.13, https://github.com/statgen/bamUtil) to merge the BAM files.

When searching for relevant germline variants, we examined 61 genes from the BROCA Cancer Risk Panel (http://tests.labmed.washington.edu/BROCA) [41, 42]. We extracted genomic data for these genes using bedtools (intersectBed module, version 2) [43].

We used Picard Tools (CalculateHsMetrics module) to calculate alignment metrics. For exome-capture regions across all germline samples, the average sequencing coverage was 44.4. The average percentage of target bases that achieved at least 30X coverage was 33.7%. The average percentage of target bases that achieved at least 100X coverage was 12.3%.

To call DNA variants, we used freebayes (version v0.9.21-18-gc15a283) [44] and Pindel (https://github.com/genome/pindel). We used freebayes to identify single-nucleotide variants (SNVs) and small insertions or deletions (indels); we used Pindel to identify medium-sized insertions and deletions. Having called these variants, we used snpEff (version 4.1) [45] to annotate the variants and GEMINI (version 0.16.3) [46] to query the variant data.

To expedite execution of the above steps, we used the GNU Parallel software [47]. The scripts and code that we used to process the germline data can be found in an open-access repository: https://bitbucket.org/srp33/tcga_germline.

Geneticists experienced in variant interpretation (BHS, TW, SG, MCK) evaluated each of the germline variants for pathogenicity. Following accepted guidelines for variant classification [42], variants were reviewed individually, followed by a group discussion. In evaluating each variant, we first considered the likelihood of pathogenicity as reported in the University of Washington, Department of Laboratory Medicine clinical-variant database. In addition, we used functional annotations from SIFT [48], Polyphen2 [49], and GERP [50]. When evaluating splice-site variants, we assessed pathogenicity based on whether the variants had been shown experimentally to cause truncations. Additionally, we used NNsplice [51] and Rescue ESE [52] to evaluate splicing variants. We also used maximum entropy modeling [53] and Human

Splicing Finder [54], which aggregate data from other splice prediction tools. We classified as benign any in-frame deletions that had been observed as naturally occurring transcripts. When evaluating potential effects of a variant on protein function, we evaluated the extent to which a given protein domain varies within the general population. This process for evaluating candidate cancer-predisposing variants is used in clinical practice and has been demonstrated to maximize actionable results and minimize the frequency of variants of unknown significance [42]. Personal and family histories were unavailable for TCGA patients, so this information could not be included in the evaluation process. The germline calls that we made for these patients were independent of variant-classification calls used in prior studies of TCGA data [25, 55].

To assess loss of heterozygosity (LOH), we used data from Riaz et al. [56]. They had made LOH calls for a large proportion of TCGA breast-cancer patients. Their process included an evaluation of data from Affymetrix SNP 6.0 arrays, genotyping via Affymetrix's birdseed algorithm, and calculation of the log ratios and B-allele frequencies using PennCNV [57]. The ASCAT algorithm [58] was used to determine allele-specific copy number and loss of heterozygosity for each mutation.

We identified somatic SNVs and indels for each patient by examining variant calls that had been made using Mutect [59]; these variants had been made available via the Genomic Data Commons [60]. Somatic variants that 1) were synonymous 2) snpEff classified as having a "LOW" or "MODIFIER" effect on protein sequence, 3) SIFT [48] and Polyphen2 [49] both suggested to be benign [61], and 4) were observed at greater than 1% frequency across all populations in ExAC [62] were excluded. For *BRCA1* and *BRCA2*, we examined candidate variants based on a priori observations of pathogenicity as reported in the University of Washington, Department of Laboratory Medicine clinical-variant database [63]. Based on these criteria, we categorized each variant as pathogenic, likely pathogenic, variant of uncertain significance (VUS), likely benign, or benign. Then we examined ClinVar [64] for evidence that VUS or likely benign variants had been classified by others as pathogenic; however, none met this criterion. To err on the side of sensitivity, we considered any *BRCA1* and *BRCA2* mutation to be "aberrant" if it fell into our pathogenic, likely pathogenic, or VUS categories. The final classifications are shown in S1 Table.

Using the somatic-mutation data for each patient, we derived mutation-signature profiles using the deconstructSigs (version 1.8.0) R package [65]. As input to this process, we used somatic-variant calls that had not been filtered for pathogenicity, as a way to ensure adequate representation of each signature. The output of this process was a vector for each tumor that indicated a weight for each signature [19]. S1 and S2 Figs illustrate these weights for two tumors that we analyzed.

We downloaded DNA methylation data via the Xena Functional Genomics Explorer [66]. These data were generated using the Illumina HumanMethylation27 and HumanMethylation450 BeadChip platforms. For the HumanMethylation27 arrays, we mapped probes to genes using a file provided by the manufacturer (https://www.ncbi.nlm.nih.gov/geo/query/acc.cgi?acc=GPL8490). For the HumanMethylation450 arrays, we mapped probes to genes using an annotation file created by Price, et al. [67] (see http://www.ncbi.nlm.nih.gov/geo/query/acc.cgi?acc=GPL16304). Typically, multiple probes mapped to a given gene. Using probe-level data from *BRCA1*, *BRCA2*, *PTEN*, and *RAD51C*, we performed a preliminary analysis to determine criteria for selecting and summarizing these probe-level values. We started with the assumption that in most cases, the genes would be methylated at low levels. We also assumed that probes nearest the transcription start site would be most informative. Upon plotting the data (S3 Fig), we decided to limit our analysis to probes that mapped to the genome within 300 nucleotides of each gene's transcription start site. In some cases, probes appeared to be faulty because they showed considerably different methylation levels ("beta" values) than other

probes in the region (S3 Fig). To mitigate the effects of these outliers, we calculated gene-level methylation values as the median beta value across any remaining probes for that gene.

To identify tumors that exhibited relatively high beta values—and thus could be considered to be hypermethylated—we used a univariate, outlier-detection algorithm, implemented in the extremevalues R packages (version 2.3.2) [68]. This enabled us to look for extreme values using one side of a specified distribution. This package supports five options for the distribution: normal (default), lognormal, exponential, pareto, and weibull. None of these distributions was a consistently good fit for the methylation (beta) values, in part because the shape of the data differed considerably across the genes (S4–S7 Figs). We used the exponential distribution because it identified hypermethylated genes at rates that were largely consistent with prior work [21]. We used the "getOutliersII" function with default parameter values.

We downloaded copy-number-variation data from the Xena Functional Genomics Explorer [66]. These data had been generated using Affymetrix SNP 6.0 arrays; CNV calls had been made using the GISTIC2 method [69]. The CNV calls had also been summarized to gene-level values using integer-based discretization. We focused on tumors with a gene count of "-2", which indicates a homozygous deletion.

We used RNA-Sequencing data that had been aligned and summarized to gene-level values using the original TCGA pipeline [25]. To facilitate biological and clinical interpretation, we examined relationships between germline and somatic events and the Prosigna™ Breast Cancer Prognostic Gene Signature (PAM50) subtypes [70]. Netanely, et al. had previously published PAM50 subtypes for TCGA breast cancer samples; we reused this information in our study [71]. We also sought to identify tumors with unusually low expression levels. To do this, we used the getOutliersI function in the extremevalues package. We used the following non-default parameter values: alpha = c(0.000001, 0.000001), distribution = "lognormal", FLim = c (0.1, 0.9). RNA-Sequencing data have been shown to fit the log-normal distribution [72].

We parsed demographic, histopathological, and surgical variables for TCGA samples from the repository prepared by Rahman, et al. [73]. We obtained drug-response data from the TCGA legacy archive (https://portal.gdc.cancer.gov/legacy-archive) and standardized drug names using synonyms from the National Cancer Institute Thesaurus [74].

## Quantitative analysis and visualization

To prepare, analyze, and visualize the data, we wrote computer scripts in the R programming language [75]. In writing these scripts, we used the following packages: readr [76], dplyr [77], ggplot2 [78], tidyr [79], reshape2 [80], ggrepel [81], cowplot [82], data.table [83], UpSetR [84], BSgenome.Hsapiens.UCSC.hg38 [85, 86], and Rtsne [87].

To reduce data dimensionality, we applied multidimensional scaling (MDS) [88] to the somatic-mutation signatures and gene-expression profiles. This reduced the data to two dimensions. To quantify homogeneity within a group of tumors that harbored a particular aberration, we calculated the pairwise Euclidean distance between each patient pair in the group and then calculated the median pairwise distance, based on the two-dimensional values [89]. As an additional measure of homogeneity, we used logistic regression to predict BRCA mutation status using the dimensionally reduced coordinates. In evaluating this approach, we used two-fold cross validation and configured the model to weight the minority class in inverse proportion to the frequency of the minority class.

When comparing two given groups, we used a similar resampling approach but instead calculated the median distance between each pair of individuals in either group. To determine whether the similarity within or between groups was statistically significant, we used a permutation approach. We randomized the patient identifiers, calculated the median pairwise

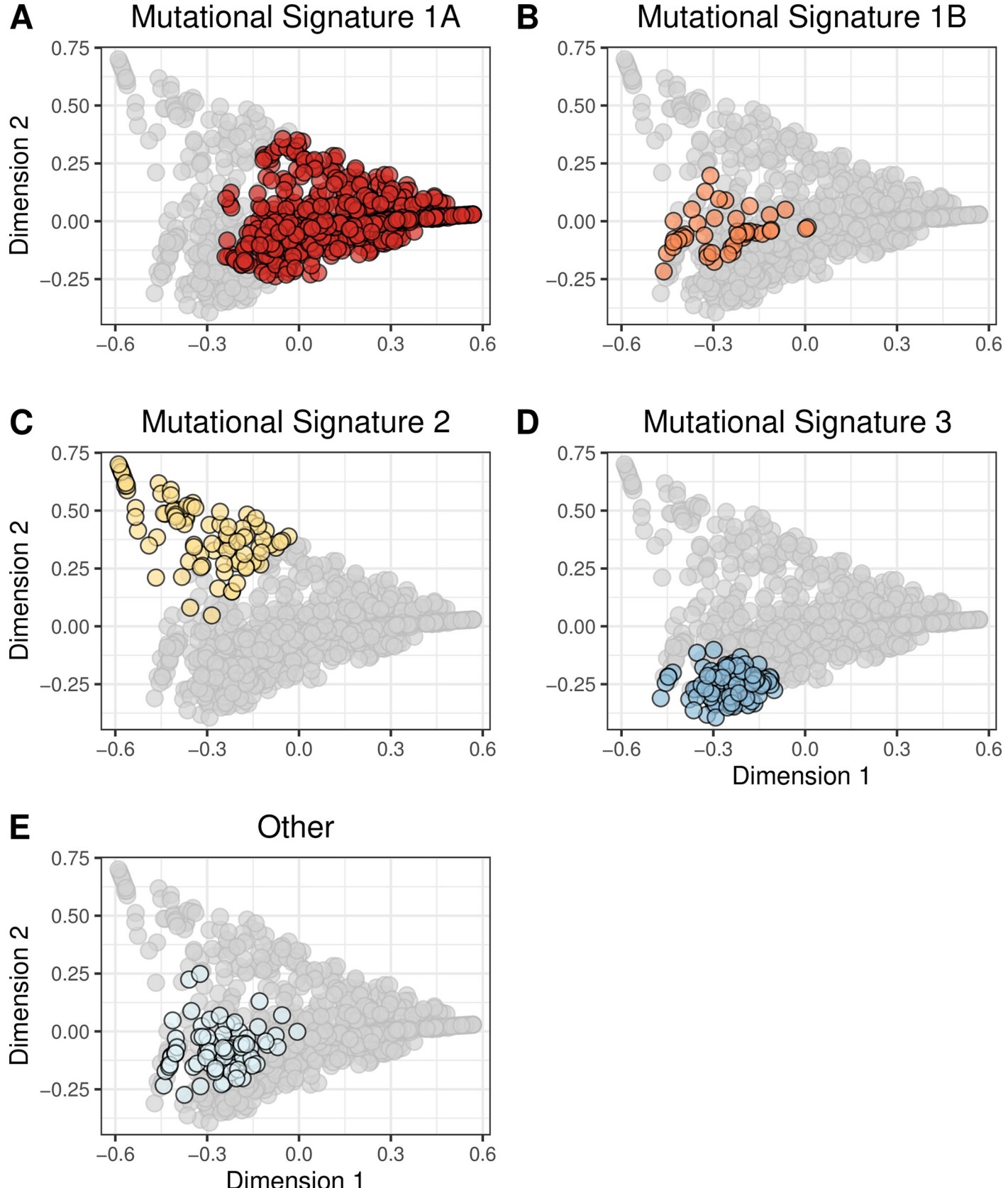

**Fig 1. Two-dimensional representation of somatic-mutation signatures using multidimensional scaling.** We summarized each tumor based on their somatic-mutation signatures, which represent overall mutational patterns in a trinucleotide context. We used multidimensional scaling (MDS) to reduce the data to two dimensions. Each point represents a single tumor, overlaid with colors that represent the tumor's primary somatic-mutation signature. Mutational Signature 1A (A) was the most prevalent; these tumors were widely dispersed across the signature landscape. Signatures 1B (B), 2 (C), and 3 (D) were relatively small and formed cohesive clusters. The remaining 23 clusters were rare individually and were dispersed broadly.

distance within (or between) groups, and repeated those steps 100,000 times. This process resulted in an empirical null distribution against which we compared the actual median distance. We then derived empirical p-values by calculating the proportion of randomized median distances that were larger than the actual median distance. We adjusted the empirical p-values for multiple testing using Holm's method [90]; we applied this correction to the p-values across all aberration types.

For visualization, we plotted the MDS values as Cartesian coordinates; we also used Barnes-Hut t-distributed Stochastic Neighbor Embedding (t-SNE) [91, 92] as an alternative method for visualizing the data.

We created a series of R scripts that execute all steps of our analysis and that generate the figures in this paper; these scripts are available at https://osf.io/9jhr2.

### Ethics statement

Brigham Young University's Institutional Review Board approved this study under exemption status. This study uses data collected from public repositories, other than the germline-variant data, which are restricted by TCGA data-access policies. We played no part in patient recruiting or in obtaining patient consent. We have adhered to guidelines from TCGA on handling data.

## Results

We used clinical and molecular data from 1101 TCGA breast-cancer patients to evaluate the downstream effects of *BRCA1* and *BRCA2* germline mutations in tumors. We evaluated two types of downstream effects: 1) signatures that reflect a tumor's overall somatic-mutation profile in a trinucleotide context and 2) tumor gene-expression levels. We used somatic-mutation signatures because they reflect the genomic effects of HR defects and have been associated with *BRCA1*/*BRCA2* mutation status [18, 19]. We used gene-expression data because they are used to classify breast tumors into subtypes [93, 94] and often reflect genomic variation [61, 95, 96]. We assessed whether either of these types of profiles was more homogeneous in *BRCA1*/*BRCA2* germline carriers than in randomly selected patients. In addition, we evaluated potential criteria for classifying tumors into the BRCA-like category. These criteria included somatic mutations, homozygous deletions, and DNA hypermethylation of *BRCA1* and *BRCA2*. Similarly, we assessed whether these different types of aberrations have downstream effects similar to *BRCA1*/*BRCA2* for 59 other cancer-predisposition genes.

### Somatic-mutation signatures and gene-expression profiles across all breast-cancer patients

First, we identified the primary somatic-mutation signature associated with each TCGA breast-cancer patient. Approximately 91% of the patients were associated with somatic-mutation signature "1A" (n = 670), "1B" (n = 48), "2" (n = 98), or "3" (n = 130). The remaining patients were assigned to 12 other signatures (S8 Fig). Second, we identified the primary PAM50 gene-expression subtype associated with each patient. The Luminal A and Luminal B subtypes were most common, but each subtype was represented by at least 37 tumors (S9 Fig).

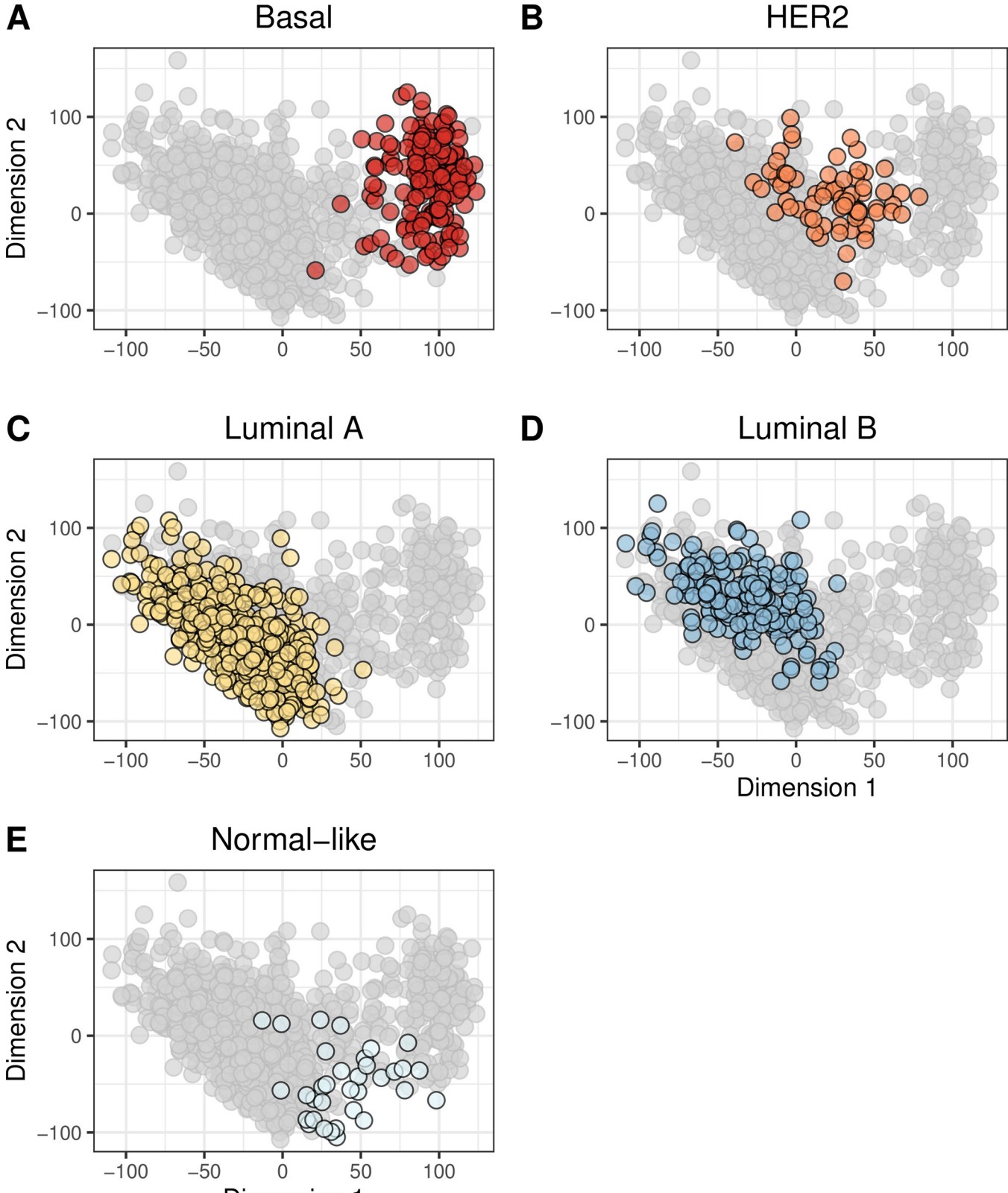

**Fig 2. Two-dimensional representation of gene-expression levels using multidimensional scaling.** We used multidimensional scaling (MDS) to reduce the gene-expression profiles to two dimensions. Each point represents a single tumor, overlaid with colors that represent the tumor's primary PAM50 subtype.

Generally, the PAM50 subtypes clustered cohesively, but there were exceptions. For example, some Basal-like tumors (A) exhbited expression patterns that differed considerably from the remaining Basal-like tumors. The normal-like tumors (E) showed the most variability in expression. This graph represents patients for whom we could identify a PAM50 subtype.

Although it is useful to evaluate the primary somatic-mutation signature or PAM50 subtype associated with each tumor, tumors are aggregates of multiple signatures and subtypes. To account for this diversity, we characterized the tumors based on 1) all 27 somatic-mutation signatures or 2) expression levels for all available genes. To enable easier interpretation of these profiles, we reduced dimensionality of the data using the MDS and t-SNE techniques (see Methods). Generally, tumors with the same primary somatic-mutation signature or PAM50 subtype clustered together in these visualizations (Figs 1 and 2; S10 and S11 Figs); however, in some cases, this did not happen. For example, the dimensionally reduced gene-expression profiles for Basal-like tumors formed a cluster that was mostly separate from the other tumors; but some Basal-like tumors were modestly distant from this cluster, and some Normal-like tumors clustered closely with the Basal-like tumors (Fig 2; S11 Fig). Tumors assigned to somatic-mutation "Signature 3" formed a cohesive cluster (Fig 1; S10 Fig), but some "Signature 3" tumors were modestly distant from this cluster. These observations highlight the importance of evaluating molecular profiles as a whole, not just using the primary category for each tumor.

## Aberrations in BRCA1 and BRCA2

Of 993 breast-cancer patients with available germline data, 22 harbored a pathogenic SNV or indel in *BRCA1*; 22 harbored a *BRCA2* variant (Fig 3A). We were able to identify loss of heterozygosity (LOH) for all but 3 *BRCA1* carriers and 7 *BRCA2* carriers (S12 and S13 Figs). A somatic mutation, homozygous deletion, or hypermethylation event occurred in *BRCA1* or *BRCA2* for 80 patients (Fig 3B–3D). Most of these events were mutually exclusive with each other and with germline variants (S14 Fig).

*BRCA1* carriers fell into the Basal (n = 17); Her2-enriched (n = 1), Luminal A (n = 2), and Luminal B (n = 1) gene-expression subtypes (S15 Fig) [22, 93, 94]. We were unable to assign a gene-expression subtype to one BRCA1 carrier due to missing data. Most *BRCA2* carriers fell into the Luminal A subtype (n = 13); the remaining individuals were dispersed across the other subtypes. As demonstrated previously [19], the primary somatic-mutation signature for most *BRCA1* and *BRCA2* carriers was "Signature 3"; however, other signatures (especially "1A") were also common (S16 Fig). S17 Fig shows the overlap between these two types of molecular profile.

## Homogeneity of somatic mutation signature and expression profiles of germline BRCA1/2 carriers

The somatic-mutation signatures of *BRCA1* germline carriers were more homogeneous than expected by chance (p = 0.00056; S18A, S19A and S20A Figs), as were those from *BRCA2* carriers (p = 0.0003; S18B, S19B and S20B Figs). As an additional measure of homogeneity, we used logistic regression to predict BRCA aberration status based on the dimensionally reduced data. Using somatic-mutation signatures, we could predict the presence of *BRCA1* germline mutations with a sensitivity of 0.72 and a specificity of 0.82. Additional classification results are shown in S2 Table.

None of the three *BRCA1* carriers who lacked LOH events clustered closely with the remaining *BRCA1* tumors (S18A and S19A Figs). Of the 7 *BRCA2* tumors without detected LOH events, 3 clustered closely with the remaining *BRCA2* tumors, while 4 did not (S18B and

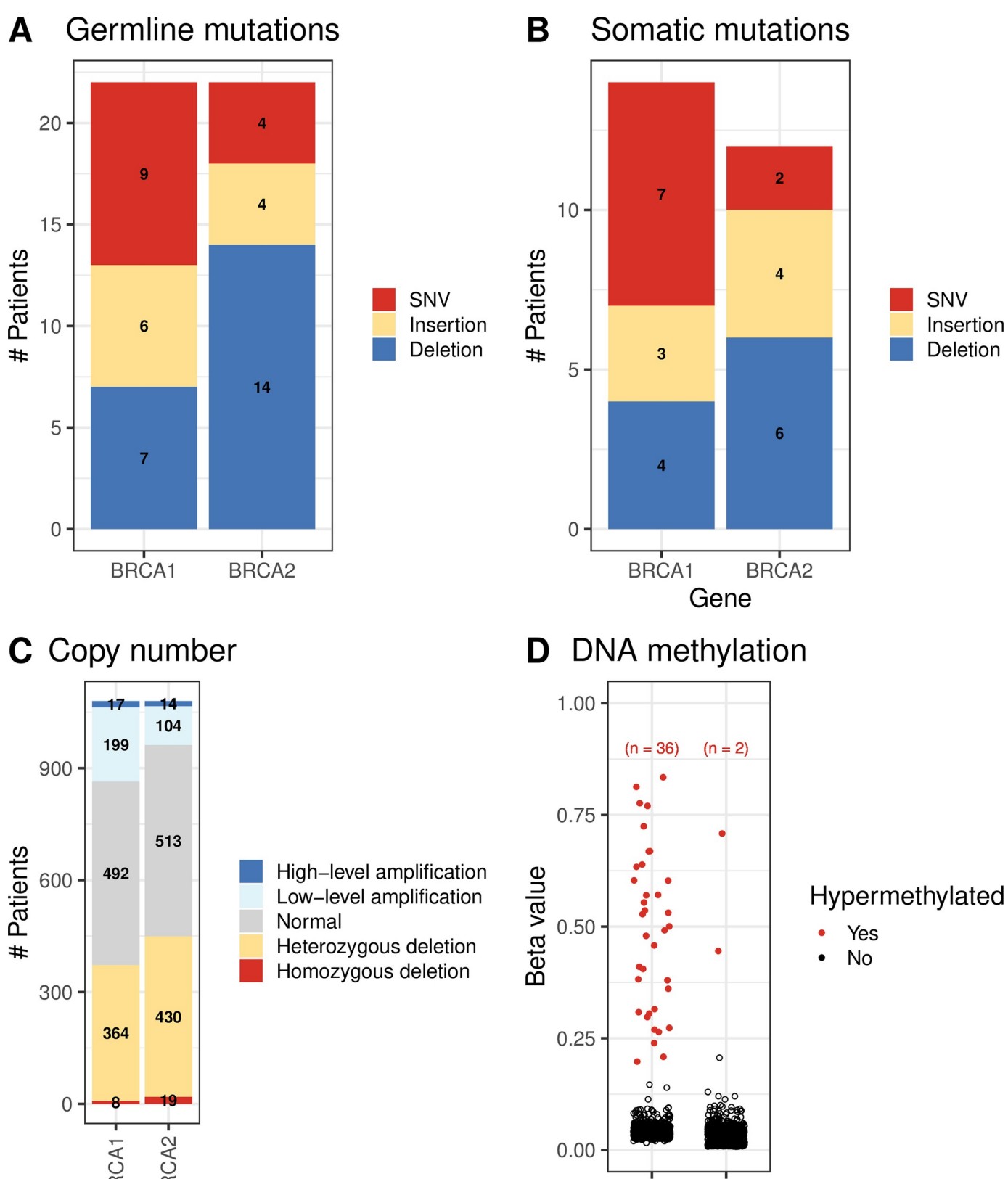

**Fig 3. Molecular aberrations in *BRCA1* and *BRCA2* across all breast-cancer patients.** A) Germline mutations, B) Somatic mutations, C) Copy-number variations, D) DNA methylation levels. SNV = single nucleotide variation.

**Table 1. Results of similarity comparisons among BRCA aberration groups.** We compared somatic-mutation signatures or gene-expression profiles between groups of patients who harbored aberrations in *BRCA1* or *BRCA2*. We evaluated whether patients in one group (e.g., those who harbored a *BRCA1* germline mutation) were more similar to patients in a second group (e.g., those with *BRCA2* germline mutation) than random patient subsets of the same sizes. The numbers in this table represent empirical p-values from our resampling approach. In cases where an individual harbored an aberration in both comparison groups, we excluded that patient from the comparison. We used Holm's method to correct for testing multiple hypotheses.

| Aberration Type 1 | Aberration Type 2 | Gene Expression | Mutational Signatures |
|---|---|---|---|
| BRCA1 germline mutation (n = 22) | BRCA2 germline mutation (n = 22) | 1.0 | 1.4e-04 |
| BRCA1 germline mutation (n = 22) | BRCA1 somatic mutation (n = 14) | 1.0 | 1.4e-04 |
| BRCA1 germline mutation (n = 22) | BRCA1 homozygous deletion (n = 8) | 1.0 | 0.30 |
| BRCA1 germline mutation (n = 22) | BRCA1 hypermethylation (n = 36) | 0.013 | 1.4e-04 |
| BRCA2 germline mutation (n = 22) | BRCA2 somatic mutation (n = 12) | 1.0 | 1.4e-04 |
| BRCA2 germline mutation (n = 22) | BRCA2 homozygous deletion (n = 19) | 1.0 | 1.4e-04 |
| BRCA2 germline mutation (n = 22) | BRCA2 hypermethylation (n = 2) | 1.0 | 5.4e-04 |

S19B Figs). It has been shown previously that germline *BRCA1/BRCA2* mutations leave a recognizable imprint on a tumor's mutational landscape [19]. This effect may be more consistent when a LOH event has occurred as a second "hit" within the same gene [97], but this was difficult to confirm with the available sample sizes.

Under the assumption that *BRCA1/BRCA2* germline variants induce recognizable effects on tumor transcription, we assessed whether tumors from *BRCA1/BRCA2* carriers have homogeneous gene-expression profiles. As expected based on the tumors' primary PAM50 classification, 17 of 21 *BRCA1* carriers (for whom we had gene-expression data) overlapped closely with the Basal-like subtype (S15 Fig). As a whole, the expression profiles for this group were more homogeneous than expected by chance (p = 0.0318; S21A, S22A and S23A Figs). However, expression values for *BRCA2* carriers were not significantly homogeneous (p = 1.0; S23B Fig). Tumors from these individuals were dispersed across the gene-expression landscape (S21B and S22B Figs).

## Similarities among individuals with *BRCA1/BRCA2* aberrations

Next we evaluated similarities between *BRCA1* and *BRCA2* germline carriers. Somatic-mutation signatures for *BRCA1* and *BRCA2* carriers were highly similar to each other (p = 0.00014; S18A, S18B, S19A, S19B and S24A Figs). However, this pattern did not hold for gene-expression profiles. Although some *BRCA2* carriers fell into the Basal-like gene-expression subtype, overall profiles for these patients were dissimilar to those from *BRCA1* carriers (p = 1.0; S21A, S21B, S22A, S22B and S25A Figs).

Whether for somatic-mutation signatures or gene-expression profiles, tumors with *BRCA1* hypermethylation were relatively homogeneous and highly similar to tumors from *BRCA1* germline carriers (Table 1; S18G, S19G, S20G, S21G, S22G and S23G Figs). For gene-expression data, no other aberration type showed significant similarity to *BRCA1* germline mutations. Somatic-mutation signatures from tumors with *BRCA1* somatic mutations were significantly similar to those from *BRCA1* germline mutations (Table 1). Only 2 tumors had *BRCA2* hypermethylation, but the mutational signatures for these samples were significantly similar to tumors from *BRCA2* germline carriers (p = 0.00054; S19H Fig). Likewise, tumors with a *BRCA2* somatic mutation or homozygous deletion had mutational signatures that were similar to germline *BRCA2* carriers (Table 1; S18D, S18F, S19D and S19F Figs). Aberrations in *BRCA1* and *BRCA2* appear to induce similar effects on somatic-mutation signatures—but not necessarily gene expression—whether those disruptions originate in the germline or via somatic events.

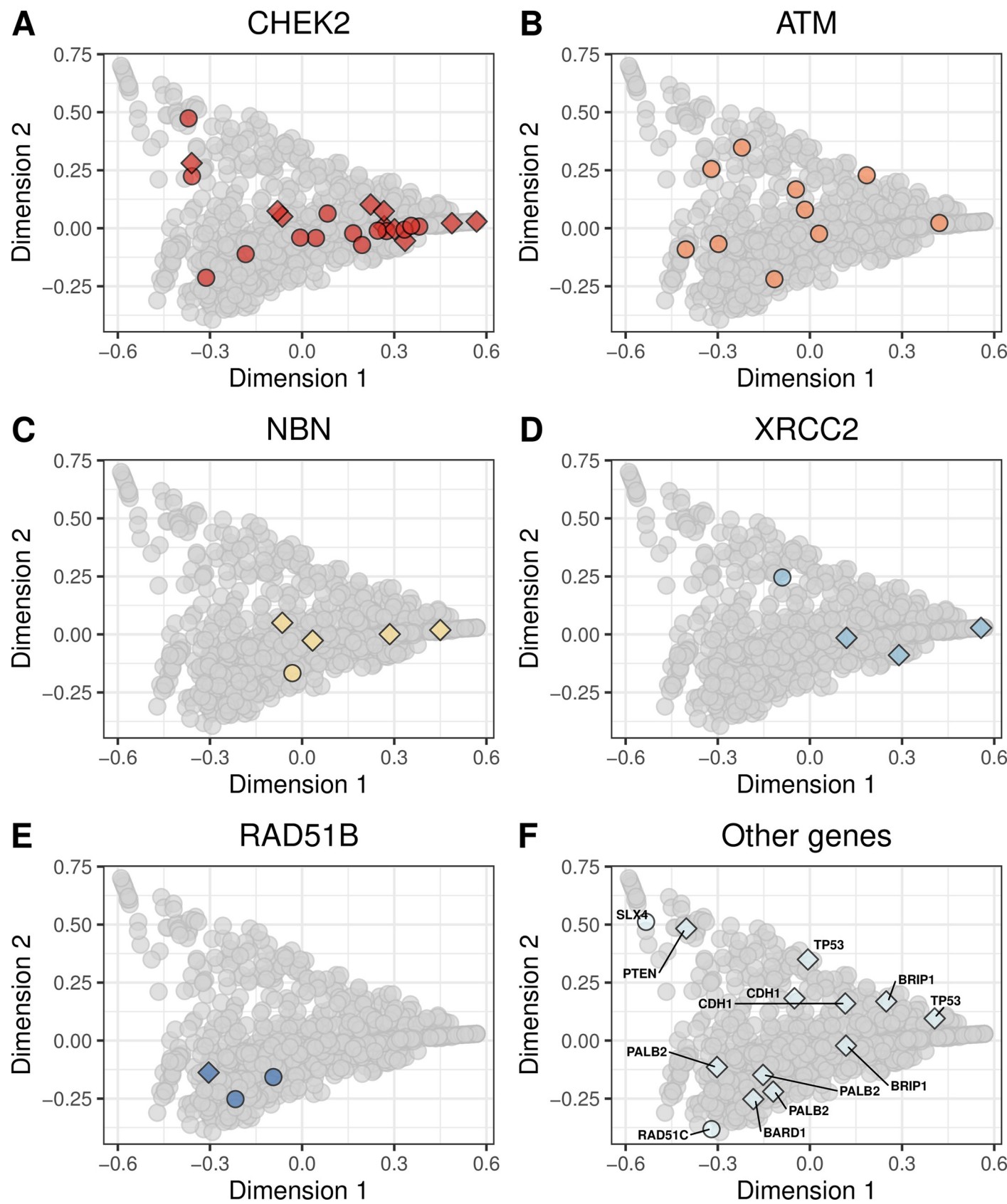

**Fig 4. Non-BRCA germline mutations on the somatic-mutation signature landscape using multidimensional scaling.** Using the same two-dimensional representation of mutational signatures shown in Fig 1, this plot indicates which patients had germline mutations in non-BRCA cancer-predisposition genes. Diamond shapes indicate patients for whom *no* loss-of-heterozygosity was observed.

## Evaluation of aberrations in other cancer-predisposing genes and clinical factors

Next we aggregated all patients who had any type of *BRCA1* or *BRCA2* aberration into a "BRCA-like reference group." As a whole, mutational signatures for this group were more homogeneous than expected by chance (p = 0.00001; S26 Fig). We used this reference group to evaluate 59 other cancer-predisposition genes that might be associated with being BRCA-like. For the remaining evaluations, we used somatic-mutation signatures only.

We evaluated whether molecular aberrations in the cancer-predisposition genes resulted in mutational signatures that were similar to our BRCA-like reference group. We found pathogenic and likely pathogenic germline mutations in 13 genes (*ATM*, *BARD1*, *BRCA1*, *BRCA2*, *BRIP1*, *CDH1*, *CHEK2*, *NBN*, *PALB2*, *PTEN*, *RAD51B*, *RAD51C*, *SLX4*, *TP53*, *XRCC2*). The most frequently mutated were *CHEK2*, *ATM*, and *NBN* (Fig 4; S27 and S28 Figs). We found potentially pathogenic somatic mutations in 55 genes, most frequently in *TP53*, *PIK3CA*, *CDH1*, and *PTEN* (Fig 5; S29 and S30 Figs). Homozygous deletions occurred most frequently in *PTEN*, *CDKN2A*, *RB1*, and *CDH1* (Fig 6; S31 and S32 Figs). Hypermethylation occurred in 22 genes, most commonly *GALNT12*, *PTCH1*, *CDKN2A*, and *RAD51C* (Fig 7, S33 and S34 Figs). Using our resampling approach, we compared each aberration type in each gene against the BRCA-like reference group. In cases where an aberration overlapped between the reference and comparison groups, we excluded individuals who harbored both aberrations. For 11 genes (*ATR*, *BARD1*, *CDKN2A*, *CTNNA1*, *PALB2*, *PALLD*, *PRSS1*, *RAD51B*, *SDHC*, *SMARCA4*, *VHL*), at least one type of aberration attained statistical significance after multiple-testing correction (Table 2). A total of 8 aberrations occurred in *BARD1*: a germline mutation, 2 somatic mutations, and 5 homozygous deletions; as a group, these tumors were statistically similar to the BRCA-like reference group (p = 0.0018). *ATR* was mutated in 15 tumors and hypermethylated in 1 tumor; together, these tumors were statistically similar to the BRCA-like reference group (p = 0.0035). Tumors with an aberration in *PRSS1* were also statistically similar to the BRCA-like reference group (p = 0.0069), but we only observed 2 aberrations in this gene. Other genes showed high similarity to the BRCA-like reference group for one type of aberration only; these included homozygous deletions in *CDKN2A* (n = 47; p = 0.00001), homozygous deletions in *CTNNA1* (n = 6; p = 0.00004), germline mutations in *PALB2* (n = 3; p = 0.00001), and homozygous deletions in *PALLD* (n = 9; p = 0.00001).

Lastly, we evaluated whether the following data types were correlated with being BRCA-like: 1) unusually low mRNA expression in a given gene, 2) demographic, histopathological, and surgical observations, and 3) patient drug responses. First, we calculated the median Euclidean distance—based on somatic-mutation signatures—between each patient and the BRCA-like reference group. Then we used a two-sided Pearson correlation test to assess the relationship between these median distances and each candidate variable. In determining whether a tumor exhibited unusually low mRNA expression for a given gene, we used an outlier-detection technique (see Methods). Unusually low expression of *BRCA1* (rho = 0.22, p = 0.0024) and *RAD51C* (rho = 0.20, p = 0.016) showed the strongest positive correlation with the reference group, whereas *CDH1* (rho = -0.19, p = 0.023), *PIK3CA* (rho = -0.19, p = 0.025) and *BARD1* (rho = -0.19, p = 0.028) showed the strongest negative correlation (S35 and S36

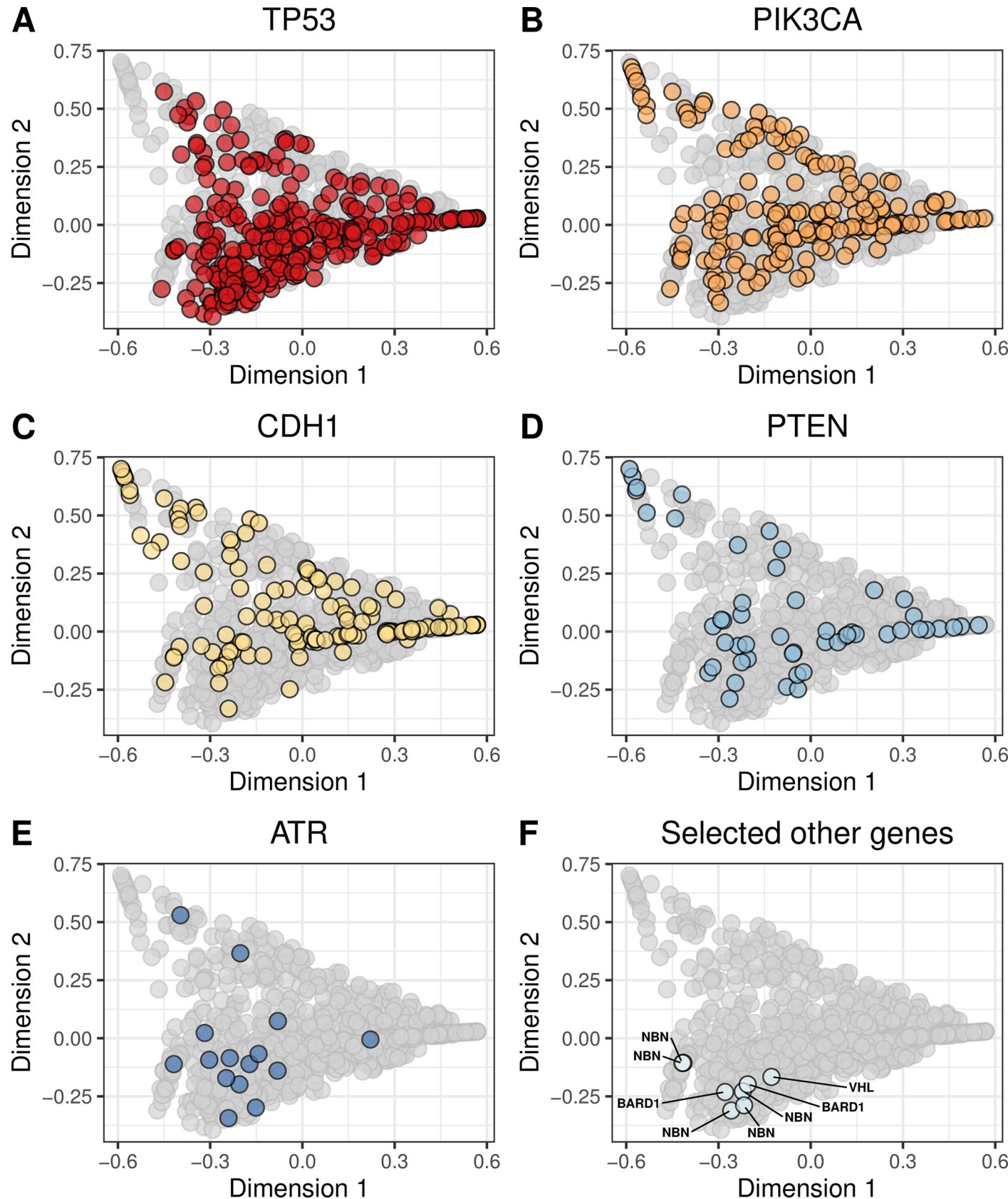

**Fig 5. Non-BRCA somatic mutations on the somatic-mutation signature landscape using multidimensional scaling.** Using the same two-dimensional representation of mutational signatures shown in Fig 1, this plot indicates which patients had somatic mutations in non-BRCA cancer-predisposition genes. Diamond shapes indicate patients for whom *no* loss-of-heterozygosity was observed.

Figs). Triple-negative status and infiltrating ductal carcinoma histology were the most positively correlated clinical variables (S37 Fig). No chemotherapy treatment was significantly associated with being BRCA-like, though availability was limited for the drug data (n = 211; S38 Fig). Relationships among low mRNA expression and these factors are likely interwoven. For example, CDH1 expression has been associated with molecular subtypes such as triple-negative status [98]. Larger studies will be necessary to disentangle these effects.

## Discussion

The concept of being BRCA-like has traditionally focused on tumors aberrations, under the hypothesis that their effects are similar to those of germline *BRCA1* and *BRCA2* mutations [31]. We evaluated this hypothesis for somatic mutations, homozygous deletions, and hypermethylation events in *BRCA1* and *BRCA2*. Corroborating prior evidence [21, 33], we found that tumors with these aberration types had somatic-mutation signatures that were similar to those of germline carriers. Using this group as a reference, we evaluated each aberration type, as well as germline mutations, in 59 other cancer-predisposition genes in search of additional criteria that might be indicators of being BRCA-like. This search identified previously known associations with being BRCA-like, including germline mutations in *PALB2* [21] and somatic mutations in ATR [32]. However, for a gene to be considered a strong candidate for inclusion in the BRCA-like definition, we required evidence of similarity across all available molecular data types. *BARD1* and *ATR* met these criteria. Both genes interact directly with BRCA1 to help repair double-stranded breaks and control G1/S cell-cycle arrest [99]. Experimental evidence lends support to the hypothesis that inactivation of these genes has relevance to being BRCA-like. For example, in mice, inactivation of the Bard1 protein induces mammary tumors that are indistinguishable from tumors that result from Brca1 knock-out [100]. In triple-negative breast cancers, ATR inhibitors are highly efficient in patient-derived xenografts that have a *BRCA1* mutation or that exhibit the BRCA-like phenocopy when combined with irinotecan, a clinically approved topoisomerase 1 inhibitor that causes double-stranded breaks [101].

Interestingly, other genes known to play a role in DNA damage repair [102]–including *ATM*, *CHEK2*, and *RAD51C*—did not attain statistical significance for any aberration type. The family-wise method we used to correct for multiple testing is generally conservative [90]. Accordingly, our results likely erred on the side of specificity rather than sensitivity for estimating whether a given gene should be considered a candidate for the BRCA-like category. This may explain why *RAD51C* hypermethylation, for example, did not reach statistical significance in our analysis, even though it has been highlighted in other studies [21, 33].

When using gene-expression profiles to characterize being BRCA-like, we observed a clear relationship between *BRCA1* aberrations and the "Basal-like" gene-expression subtype, confirming prior findings [22–25]. Our findings extended to triple-negative tumors, again confirming prior evidence [26, 27]. Hedenfalk, et al. demonstrated that breast tumors from *BRCA1* and *BRCA2* carriers exhibit gene-expression patterns that are distinct from each other [103]; our analysis confirmed these findings. Rice, et al. identified correlations between transcriptional inactivation of *BRCA1* and either germline mutations or hypermethylation of the same gene [104]. Considering potential effects across the transcriptome, our analysis provided additional evidence that gene-expression profiles from tumors of *BRCA1* germline carriers are highly similar to tumors with *BRCA1* hypermethylation. These findings did not extend to

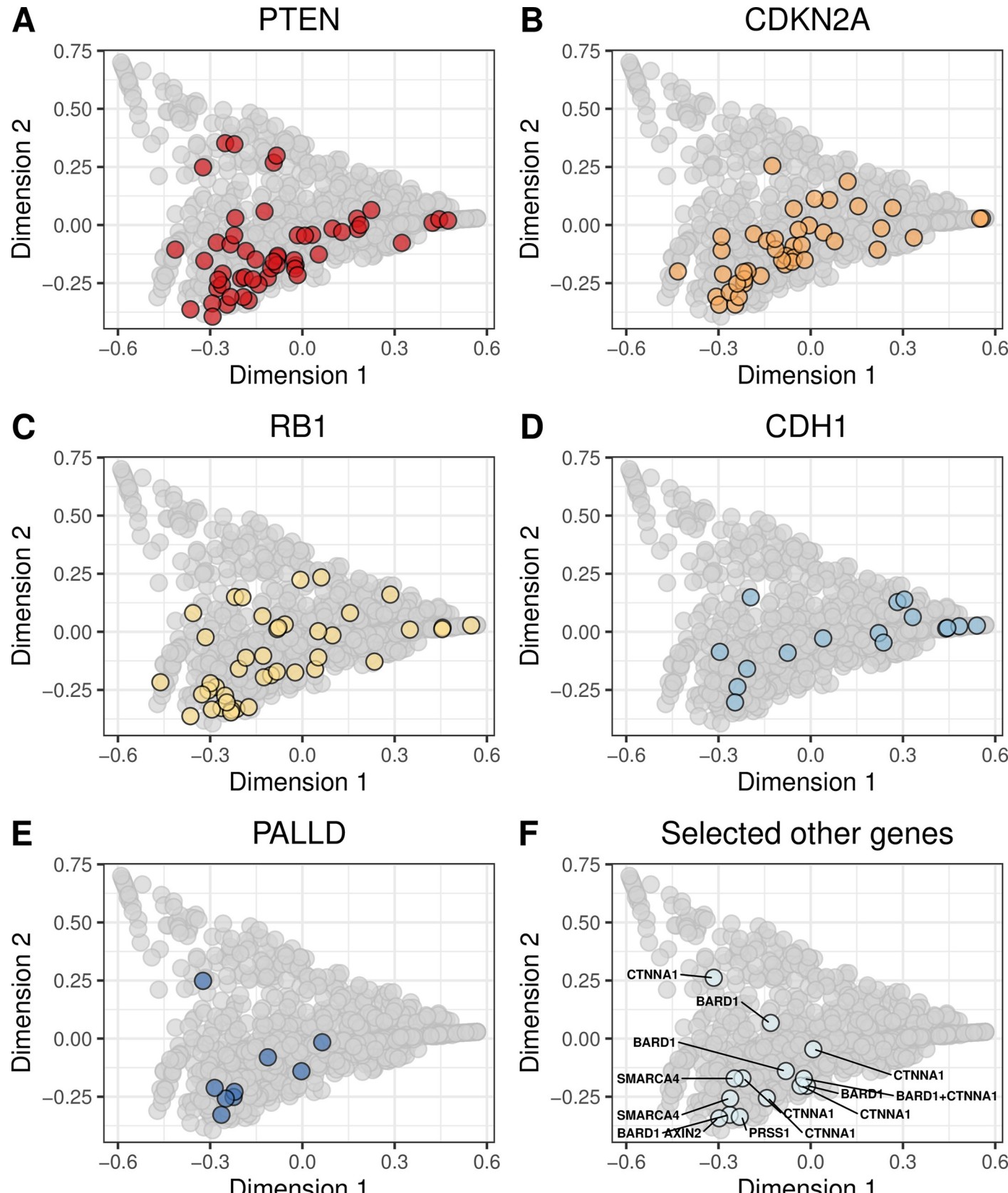

**Fig 6. Non-BRCA homozygous deletions on the somatic-mutation signature landscape using multidimensional scaling.** Using the same two-dimensional representation of mutational signatures shown in Fig 1, this plot indicates which patients had homozygous deletions in non-BRCA cancer-predisposition genes. Diamond shapes indicate patients for whom *no* loss-of-heterozygosity was observed.

*BRCA2*. Moelans, et al. found that *BRCA2* hypermethylation occurs frequently in ductal carcinoma in situ lesions and in adjacent invasive ductal cancer cells [105]; however, we identified only 2 samples that were hypermethylated for this gene.

Although gene-expression data may be less useful than somatic-mutation signatures for characterizing the effects of HR defects, they may hold promise as predictive biomarkers for specific patient subgroups. For example, Tutt, et al. observed that triple-negative hormone status was a reliable biomarker of objective treatment responses to carboplatin [106]. Severson, et al. treated HER2-negative patients with a combination of veliparib, a PARP inhibitor, and carboplatin, a platinum agent, in a neoadjuvant setting. In addition, they derived a gene-expression signature that characterized "BRCA1ness" and found that this signature was associated with response to this combination therapy [28]. In ovarian tumors, Konstantinopoulos, et al. derived a gene-expression signature that distinguished "BRCA-like" from "non–BRCA-like" samples and used this signature to accurately predict sensitivity or resistance to a platinum agent and a PARP inhibitor in patient-derived specimens [29]. Finally, Mulligan, et al. developed a 44-gene assay and showed that differences in expression of these genes between *BRCA1*/*BRCA2* mutant and sporadic tumors were predictive of response to DNA-damaging agents [107]. We attempted to replicate these findings with TCGA data, but sample sizes were too small on a per-drug basis.

Davies, et al. aimed to develop a BRCA-like biomarker using somatic-mutation signatures [33]. Using a lasso logistic-regression model, they were able to identify aberrant *BRCA1*/*BRCA2* tumors with near perfect accuracy and then extended their approach to identify tumors with an apparent functional HR deficiency that lacked known aberrations in these genes. In contrast, we focused on identifying candidate BRCA-like genes, rather than to develop a predictive biomarker for being BRCA-like itself (or for treatment responses). By identifying such genes, we aimed to provide insights into possible mechanisms of being BRCA-like. Such insights might indirectly be useful for predicting sensitivity to PARP inhibitors or platinum agents, although this connection is still tenuous [32, 108]. Polak, et al. used an alternative methodology to ours, associating somatic-mutation signatures with genomic aberrations in TCGA breast-cancer samples; they identified some of the same relationships that we identified [21]. However, our analysis extended to additional genes and factors, including extremely low-expressing genes and clinical variables. In addition, we explicitly compared the effects on somatic-mutation signatures of *BRCA1* versus *BRCA2* aberrations.

Different types of aberration may result in different downstream effects; however, these differences may result from technical challenges in identifying and filtering variants. Determining which genmic aberrations are pathogenic remains a challenging task [109], so it is likely that more- or less-stringent filtering of candidate aberrations would lead to more accurate results. In addition, we could not always determine whether mono- or bi-allelic inactivation of a given gene had occurred in a given tumor; mono-allelic inactivation may be insufficient to impair HR function [56].

Finally, we note methodological issues. To enable easier visualization, we reduced the molecular signatures to two dimensions. We also used the dimensionally reduced data as input to our statistical resampling approach. Accordingly, even though the number of original input variables was much larger for the gene-expression data, this approach enabled us to

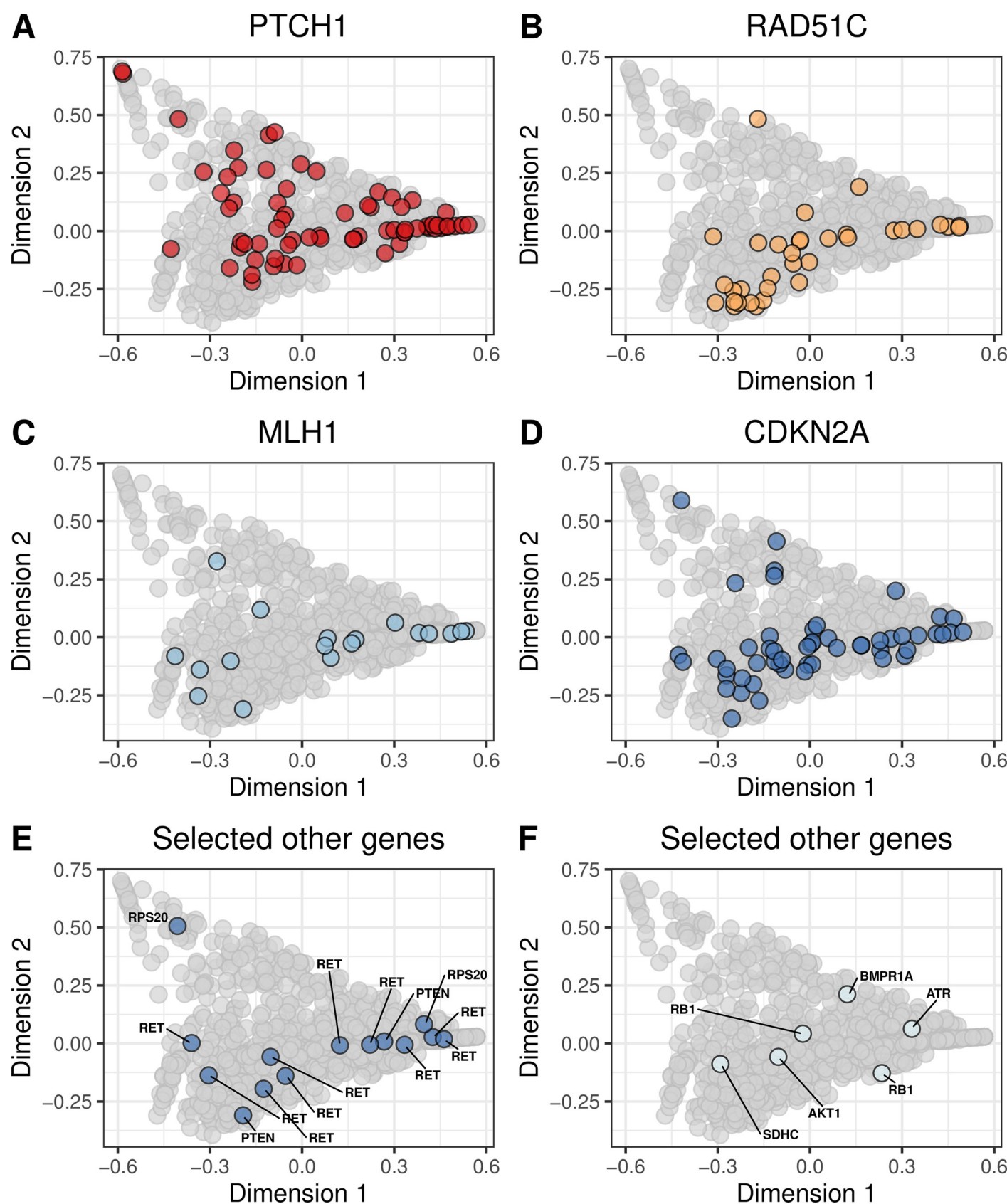

**Fig 7. Non-BRCA hypermethylation events on the somatic-mutation signature landscape using multidimensional scaling.** Using the same two-dimensional representation of mutational signatures shown in Fig 1, this plot indicates which patients had hypermethylation events in non-BRCA cancer-predisposition genes. Diamond shapes indicate patients for whom *no* loss-of-heterozygosity was observed.

perform a more consistent comparison between the two data types. However, reducing the data to this extent likely failed to capture much of the biological signal in the data for either data type. Further refining this approach may help to strike a better balance between data interpretability and adequate data representation [110].

**Table 2. Summary of comparisons between the BRCA-like reference group and groups of patients who harbored a specific type of aberration in a candidate BRCA-like gene.** We evaluated whether somatic-mutation signatures from patients who harbored a given type of aberration (e.g., BARD1 germline mutation) were more similar to the BRCA-like reference group than expected by random chance. The numbers in this table represent empirical p-values from our resampling approach. In cases where no patient had a given type of aberration in a given gene, we list "N/A". The "Any" group represents individuals who harbored any type of aberration in a given gene. We used Holm's method to correct for testing multiple hypotheses.

| Gene | Germline mutation | Somatic mutation | Homozygous deletion | Hypermethylation | Any |
|---|---|---|---|---|---|
| AKT1 | N/A | 1.0 (n = 21) | 1.0 (n = 3) | 0.13 (n = 1) | 1.0 (n = 25) |
| APC | N/A | 1.0 (n = 19) | 1.0 (n = 7) | N/A | 1.0 (n = 26) |
| ATM | 1.0 (n = 11) | 1.0 (n = 18) | 1.0 (n = 16) | N/A | 1.0 (n = 46) |
| ATR | N/A | 0.0018 (n = 15) | N/A | 1.0 (n = 1) | 0.0035 (n = 16) |
| AXIN2 | N/A | 1.0 (n = 6) | N/A | N/A | 1.0 (n = 7) |
| BAP1 | N/A | 1.0 (n = 6) | 1.0 (n = 7) | N/A | 1.0 (n = 13) |
| BARD1 | 0.0018 (n = 1) | 0.0018 (n = 2) | 0.054 (n = 5) | N/A | 0.0018 (n = 8) |
| BMPR1A | N/A | 1.0 (n = 2) | 1.0 (n = 8) | 1.0 (n = 1) | 1.0 (n = 11) |
| BRIP1 | 1.0 (n = 2) | 1.0 (n = 8) | N/A | N/A | 1.0 (n = 11) |
| CDH1 | 1.0 (n = 2) | 1.0 (n = 135) | 1.0 (n = 22) | N/A | 1.0 (n = 159) |
| CDK4 | N/A | 1.0 (n = 2) | N/A | N/A | 1.0 (n = 3) |
| CDKN2A | N/A | N/A | 0.0018 (n = 47) | 1.0 (n = 53) | 0.57 (n = 100) |
| CHEK1 | N/A | 1.0 (n = 4) | 1.0 (n = 16) | N/A | 1.0 (n = 20) |
| CHEK2 | 1.0 (n = 25) | 1.0 (n = 6) | 1.0 (n = 1) | N/A | 1.0 (n = 32) |
| CTNNA1 | N/A | 1.0 (n = 8) | 0.0069 (n = 6) | N/A | 1.0 (n = 14) |
| FAM175A | N/A | 0.085 (n = 2) | 1.0 (n = 3) | N/A | 0.055 (n = 5) |
| FH | N/A | 1.0 (n = 3) | 1.0 (n = 1) | N/A | 1.0 (n = 4) |
| FLCN | N/A | 1.0 (n = 2) | 1.0 (n = 8) | N/A | 1.0 (n = 10) |
| GALNT12 | N/A | 1.0 (n = 7) | 1.0 (n = 3) | 1.0 (n = 79) | 1.0 (n = 89) |
| GEN1 | N/A | 1.0 (n = 2) | 1.0 (n = 3) | N/A | 1.0 (n = 5) |
| GREM1 | N/A | 1.0 (n = 2) | 1.0 (n = 15) | N/A | 1.0 (n = 17) |
| HOXB13 | N/A | 1.0 (n = 3) | N/A | N/A | 1.0 (n = 4) |
| MEN1 | N/A | 1.0 (n = 9) | 1.0 (n = 3) | N/A | 1.0 (n = 13) |
| MLH1 | N/A | 1.0 (n = 13) | 1.0 (n = 4) | 1.0 (n = 19) | 1.0 (n = 36) |
| MRE11A | N/A | 1.0 (n = 2) | 1.0 (n = 8) | 1.0 (n = 2) | 1.0 (n = 12) |
| MSH2 | N/A | 1.0 (n = 4) | N/A | N/A | 1.0 (n = 5) |
| MSH6 | N/A | 1.0 (n = 7) | N/A | N/A | 1.0 (n = 8) |
| MUTYH | N/A | 1.0 (n = 3) | N/A | N/A | 1.0 (n = 3) |
| NBN | 1.0 (n = 5) | 1.0 (n = 5) | N/A | N/A | 1.0 (n = 10) |
| NF1 | N/A | 1.0 (n = 40) | 1.0 (n = 7) | N/A | 1.0 (n = 47) |
| NTHL1 | N/A | 1.0 (n = 1) | N/A | N/A | 1.0 (n = 1) |
| PALB2 | 0.0018 (n = 3) | 1.0 (n = 5) | N/A | N/A | 0.12 (n = 8) |
| PALLD | N/A | 1.0 (n = 6) | 0.0018 (n = 9) | 1.0 (n = 7) | 1.0 (n = 22) |
| PIK3CA | N/A | 1.0 (n = 190) | N/A | N/A | 1.0 (n = 191) |
| PMS2 | N/A | 1.0 (n = 5) | 0.49 (n = 1) | N/A | 1.0 (n = 6) |

*(Continued)*

**Table 2.** (Continued)

| Gene | Germline mutation | Somatic mutation | Homozygous deletion | Hypermethylation | Any |
|------|-------------------|------------------|---------------------|------------------|-----|
| POLD1 | N/A | 1.0 (n = 3) | 1.0 (n = 2) | N/A | 1.0 (n = 6) |
| POLE | N/A | 0.091 (n = 16) | 1.0 (n = 4) | N/A | 0.33 (n = 20) |
| POT1 | N/A | 1.0 (n = 5) | 1.0 (n = 1) | N/A | 1.0 (n = 6) |
| PRKAR1A | N/A | 1.0 (n = 7) | N/A | N/A | 1.0 (n = 7) |
| PRSS1 | N/A | 1.0 (n = 1) | 0.0018 (n = 1) | N/A | 0.0069 (n = 2) |
| PTCH1 | N/A | 1.0 (n = 16) | 1.0 (n = 3) | 1.0 (n = 69) | 1.0 (n = 88) |
| PTEN | 1.0 (n = 1) | 1.0 (n = 51) | 0.30 (n = 56) | 1.0 (n = 2) | 1.0 (n = 110) |
| RAD51B | 0.0018 (n = 3) | 1.0 (n = 3) | 1.0 (n = 9) | N/A | 0.64 (n = 15) |
| RAD51C | 0.61 (n = 1) | 1.0 (n = 3) | 1.0 (n = 2) | 0.098 (n = 35) | 0.61 (n = 41) |
| RAD51D | N/A | 1.0 (n = 3) | 1.0 (n = 4) | N/A | 1.0 (n = 7) |
| RB1 | N/A | 1.0 (n = 19) | 1.0 (n = 45) | 1.0 (n = 2) | 1.0 (n = 66) |
| RET | N/A | 1.0 (n = 5) | 1.0 (n = 2) | 1.0 (n = 10) | 1.0 (n = 17) |
| RINT1 | N/A | 1.0 (n = 5) | N/A | N/A | 1.0 (n = 6) |
| RPS20 | N/A | N/A | N/A | 1.0 (n = 2) | 1.0 (n = 3) |
| SDHB | N/A | 1.0 (n = 1) | 1.0 (n = 7) | N/A | 1.0 (n = 9) |
| SDHC | N/A | N/A | N/A | 0.03 (n = 1) | 0.032 (n = 1) |
| SDHD | N/A | N/A | 1.0 (n = 16) | N/A | 1.0 (n = 16) |
| SLX4 | 1.0 (n = 1) | 1.0 (n = 10) | N/A | N/A | 1.0 (n = 12) |
| SMAD4 | N/A | 1.0 (n = 12) | 1.0 (n = 14) | N/A | 1.0 (n = 26) |
| SMARCA4 | N/A | 1.0 (n = 7) | 0.0018 (n = 2) | N/A | 1.0 (n = 9) |
| STK11 | N/A | 0.13 (n = 2) | 1.0 (n = 12) | N/A | 1.0 (n = 14) |
| TP53 | 1.0 (n = 2) | 1.0 (n = 302) | 1.0 (n = 15) | N/A | 1.0 (n = 319) |
| VHL | N/A | 0.0018 (n = 1) | 1.0 (n = 1) | N/A | 1.0 (n = 2) |
| XRCC2 | 1.0 (n = 4) | 1.0 (n = 3) | 1.0 (n = 4) | N/A | 1.0 (n = 11) |

## Supporting information

**S1 Fig. Somatic-mutation signature weights for one Signature 3 tumor.** This tumor had a large proportion of C>T mutations, which are representative of Signature 3.
(PDF)

**S2 Fig. Somatic-mutation signature weights for a second Signature 3 tumor.** This tumor had a large proportion of C>T mutations, which are representative of Signature 3.
(PDF)

**S3 Fig. Probe-level summarization of DNA methylation probes.** We extracted probe-level methylation (beta) values for all available breast-cancer samples in TCGA and plotted them relative to the transcription start site of each gene. These graphs illustrate beta values for four genes (BRCA1, BRCA2, PTEN, and RAD51C) and two microarray platforms (Illumina HumanMethylation 27K and 450K). Values in parenthesis indicate distance from the transcription start site (TSS). TSS distances marked as "NA" were unavailable. The 27K arrays have fewer probes per gene. In general, probes near the TSS exhibited relatively low methylation levels for these genes, whereas probes further from the TSS were more highly methylated. These observations are consistent with the assumption that most genes would be "on" by default. Some exceptions to this pattern are apparent (for example, cg13782816 on panel C); these exceptions may be caused by mismapped probes, cross hybridization, or misannotations. We calculated gene-level values as the median across all probes that were within 300

nucleotides of the TSS.
(PDF)

**S4 Fig. Fit of outlier-detection model to DNA methylation data for BRCA1.** We used an outlier-detection methodology to estimate which tumors were hypermethylated for a given gene. This scatter plot illustrates the model fit for BRCA1. Asterisks represent tumors considered to be outliers.
(PDF)

**S5 Fig. Fit of outlier-detection model to DNA methylation data for BRCA2.** We used an outlier-detection methodology to estimate which tumors were hypermethylated for a given gene. This scatter plot illustrates the model fit for BRCA2. Asterisks represent tumors considered to be outliers.
(PDF)

**S6 Fig. Fit of outlier-detection model to DNA methylation data for PTEN.** We used an outlier-detection methodology to estimate which tumors were hypermethylated for a given gene. This scatter plot illustrates the model fit for PTEN. Asterisks represent tumors considered to be outliers.
(PDF)

**S7 Fig. Fit of outlier-detection model to DNA methylation data for RAD51C.** We used an outlier-detection methodology to estimate which tumors were hypermethylated for a given gene. This scatter plot illustrates the model fit for RAD51C. Asterisks represent tumors considered to be outliers.
(PDF)

**S8 Fig. Primary somatic-mutation signature across all breast-cancer patients.** Each TCGA breast-cancer patient was assigned to a primary somatic-mutation signature based on exome-wide mutational patterns. This plot illustrates the frequency of each somatic-mutation signature across all the patients.
(PDF)

**S9 Fig. Primary PAM50 subtypes across all breast-cancer patients.** Each TCGA breast-cancer patient was assigned to a primary PAM50 subtype based on tumor gene-expression levels. This plot illustrates the frequency of each PAM50 subtype across all the patients.
(PDF)

**S10 Fig. Two-dimensional representation of somatic-mutation signatures using the t-SNE method.** We summarized each tumor based on their somatic-mutation signatures, which represent overall mutational patterns in a trinucleotide context. We used the t-distributed Stochastic Neighbor Embedding (t-SNE) method to reduce the data to two dimensions. Each point represents a single tumor, overlaid with colors that represent the tumor's primary somatic-mutation signature. Mutational Signature 1A (A) was the most prevalent; these tumors were widely dispersed across the signature landscape. Signatures 1B (B), 2 (C), and 3 (D) were relatively small and formed cohesive clusters. The remaining 23 clusters were rare individually and were dispersed broadly.
(PDF)

**S11 Fig. Two-dimensional representation of gene-expression levels using the t-SNE method.** We used the t-distributed Stochastic Neighbor Embedding (t-SNE) method to reduce the gene-expression profiles to two dimensions. Each point represents a single tumor, overlaid with colors that represent the tumor's primary PAM50 subtype. Generally, the PAM50

subtypes clustered cohesively, but there were exceptions. For example, some Basal-like tumors (A) exhbited expression patterns that differed considerably from the remaining Basal-like tumors. The normal-like tumors (E) showed the most variability in expression. This graph represents patients for whom we could identify a PAM50 subtype.
(PDF)

**S12 Fig. Intersection between germline-mutation status and loss of heterozygosity for BRCA1.** A total of 22 patients carried a germline mutation in BRCA1. We detected loss-of-heterozygosity events in tumors for all but 3 of these patients. Data are only shown for patients for whom we had both types of data.
(PDF)

**S13 Fig. Intersection between germline-mutation status and loss of heterozygosity for BRCA2.** A total of 22 patients carried a germline mutation in BRCA2. We detected loss-of-heterozygosity events in tumors from all but 7 of these patients. Data are only shown for patients for whom we had both types of data.
(PDF)

**S14 Fig. Intersection between different types of molecular aberration in BRCA1 and BRCA2.** This graph indicates how many patients had each type of molecular aberration and the level of overlap among these aberrations within a given patient. In most cases, these aberrations were mutually exclusive from each other; however, some overlap did occur. For example, one patient had a somatic mutation in BRCA1 and hypermethylation of the same gene. This graph only depicts patients for whom all four types of molecular data were available.
(PDF)

**S15 Fig. Overlap between BRCA1/BRCA2 germline-mutation status and PAM50 subtype.** Gene-expression subtypes were unavailable for some patients; One BRCA1 carrier is not represented in this figure; we could not assign a gene-expression subtype to this individual due to missing data.
(PDF)

**S16 Fig. Overlap between BRCA1/BRCA2 germline-mutation status and primary somatic-mutation signature.** This graph represents patients for whom we had both germline- and somatic-mutation data.
(PDF)

**S17 Fig. Overlap between PAM50 subtype and primary somatic-mutation signature.** This graph represents patients for whom we could evaluate the status of both PAM50 subtype and somatic-mutation signatures.
(PDF)

**S18 Fig. BRCA1 and BRCA2 aberrations on the somatic-mutation signature landscape using multidimensional scaling.** Using the same two-dimensional representation of mutational signatures shown in Fig 1, this plot indicates which patients had germline mutations (A, B), somatic mutations (C, D), homozygous deletions (E, F), or hypermethylation events (G, H) in BRCA1 and BRCA2, respectively. Largely, these tumors had similar somatic-mutation signatures. Diamond shapes indicate patients for whom no loss-of-heterozygosity was observed. Data are shown for all patients, even those for whom we did not have all types of data.
(PDF)

**S19 Fig. BRCA1 and BRCA2 aberrations on the somatic-mutation signature landscape using the t-SNE method.** Using the same two-dimensional representation of mutational

signatures shown in S10 Fig, this plot indicates which patients had germline mutations (A, B), somatic mutations (C, D), homozygous deletions (E, F), or hypermethylation events (G, H) in BRCA1 and BRCA2, respectively. Largely, these tumors had similar somatic-mutation signatures. Diamond shapes indicate patients for whom no* loss-of-heterozygosity was observed. Data are shown for all patients, even those for whom we did not have all types of data. (PDF)

**S20 Fig. Euclidean distances for randomly selected patients compared to actual distances within BRCA1/BRCA2 patient groups based on somatic-mutation signatures.** We calculated the Euclidean distance between each pair of individuals who had germline mutations (A, B), somatic mutations (C, D), homozygous deletions (E, F), or hypermethylation events (G, H) in BRCA1 or BRCA2; the medians of these distances are illustrated using vertical, dashed lines. We then randomized the patient identifiers and calculated pairwise distances for the same number of randomly selected patients, which resulted in an empirical null distribution. We calculated p-values by comparing the actual distances against the randomized distances and then adjusted for multiple tests using Holm's method. (PDF)

**S21 Fig. BRCA1 and BRCA2 aberrations on the gene-expression landscape using multidimensional scaling.** Using the same two-dimensional representation of gene-expression profiles shown in Fig 2, this plot indicates which patients had germline mutations (A, B), somatic mutations (C, D), homozygous deletions (E, F), or hypermethylation events (G, H) in BRCA1 and BRCA2, respectively. Many of these tumors overlapped with the Basal-like subtype, but other tumors were dispersed broadly across the gene-expression landscape. Diamond shapes indicate patients for whom no loss-of-heterozygosity was observed. Data are shown for all patients, even those for whom we did not have all types of data. (PDF)

**S22 Fig. BRCA1 and BRCA2 aberrations on the gene-expression landscape using the t-SNE method.** Using the same two-dimensional representation of gene-expression profiles shown in S11 Fig, this plot indicates which patients had germline mutations (A, B), somatic mutations (C, D), homozygous deletions (E, F), or hypermethylation events (G, H) in BRCA1 and BRCA2, respectively. Many of these tumors overlapped with the Basal-like subtype, but other tumors were dispersed broadly across the gene-expression landscape. Diamond shapes indicate patients for whom no loss-of-heterozygosity was observed. Data are shown for all patients, even those for whom we did not have all types of data. (PDF)

**S23 Fig. Euclidean distances for randomly selected patients compared to actual distances within BRCA1/BRCA2 patient groups based on gene-expression profiles.** We calculated the Euclidean distance between each pair of individuals who had germline mutations (A, B), somatic mutations (C, D), homozygous deletions (E, F), or hypermethylation events (G, H) in BRCA1 or BRCA2; the medians of these distances are illustrated using vertical, dashed lines. We then randomized the patient identifiers and calculated pairwise distances for the same number of randomly selected patients, which resulted in an empirical null distribution. We calculated p-values by comparing the actual distances against the randomized distances and then adjusted for multiple tests using Holm's method. (PDF)

**S24 Fig. Somatic-mutation signature-based Euclidean distances for randomly selected patient pairs compared to actual distances between patient pairs for individuals with**

**BRCA1/BRCA2 aberrations.** We identified patients who had a germline mutation in BRCA1 or BRCA2 and compared them against each other (A), those with a somatic mutation in the same gene (B-C), those with a homozygous deletion in the same gene (D-E) and those with DNA hypermethylation of the same gene (F-G). We calculated the Euclidean distance between each pair of individuals in these groups; these distances are illustrated using vertical, dashed lines. We then randomized the patient identifiers and calculated pairwise distances for groups of randomly selected patients, which resulted in an empirical null distribution. We calculated p-values by comparing the actual distances against the randomized distances.
(PDF)

**S25 Fig. Gene-expression based Euclidean distances for randomly selected patient pairs compared to actual distances between patient pairs for individuals with BRCA1/BRCA2 aberrations.** We identified patients who had a germline mutation in BRCA1 or BRCA2 and compared them against each other (A), those with a somatic mutation in the same gene (B-C), those with a homozygous deletion in the same gene (D-E) and those with DNA hypermethylation of the same gene (F-G). We calculated the Euclidean distance between each pair of individuals in these groups; these distances are illustrated using vertical, dashed lines. We then randomized the patient identifiers and calculated pairwise distances for groups of randomly selected patients, which resulted in an empirical null distribution. We calculated p-values by comparing the actual distances against the randomized distances.
(PDF)

**S26 Fig. Euclidean distances for randomly selected patients compared to actual distances across all patients with a BRCA1 or BRCA2 aberration based on somatic-mutation signatures.** We calculated the Euclidean distance between each pair of individuals who had a germline mutation, somatic mutation, homozygous deletion, and/or hypermethylation event in BRCA1 and/or BRCA2; the median of these distances is illustrated using a vertical, dashed line. We then randomized the patient identifiers and calculated pairwise distances for the same number of randomly selected patients, which resulted in an empirical null distribution. We calculated a p-value by comparing the actual distance against the randomized distances.
(PDF)

**S27 Fig. Number of patients with germline mutations in non-BRCA cancer-predisposition genes.** This graph omits genes in which we observed no germline mutations. SNV = single-nucleotide variant.
(PDF)

**S28 Fig. Non-BRCA germline mutations on the somatic-mutation signature landscape using the t-SNE method.** Using the same two-dimensional representation of mutational signatures shown in S10 Fig, this plot indicates which patients had germline mutations in non-BRCA cancer-predisposition genes. Diamond shapes indicate patients for whom no loss-of-heterozygosity was observed.
(PDF)

**S29 Fig. Number of patients with somatic mutations in non-BRCA cancer-predisposition genes.** This graph omits genes in which we observed no somatic mutations.
(PDF)

**S30 Fig. Non-BRCA somatic mutations on the somatic-mutation signature landscape using the t-SNE method.** Using the same two-dimensional representation of mutational signatures shown in S10 Fig, this plot indicates which patients had somatic mutations in non-BRCA cancer-predisposition genes. Diamond shapes indicate patients for whom no loss-of-

heterozygosity was observed.
(PDF)

**S31 Fig. Number of patients with homozygous deletions in non-BRCA cancer-predisposition genes.** This graph omits genes in which we observed no homozygous deletions.
(PDF)

**S32 Fig. Non-BRCA homozygous deletions on the somatic-mutation signature landscape using the t-SNE method.** Using the same two-dimensional representation of mutational signatures shown in S10 Fig, this plot indicates which patients had homozygous deletions in non-BRCA cancer-predisposition genes. Diamond shapes indicate patients for whom no loss-of-heterozygosity was observed.
(PDF)

**S33 Fig. DNA methylation (beta) values for non-BRCA cancer-predisposition genes.** Tumors that we classified as having hypermethylation events are highlighted as red points. This graph omits genes in which we observed no hypermethylation events.
(PDF)

**S34 Fig. Non-BRCA hypermethylation events on the somatic-mutation signature landscape using the t-SNE method.** Using the same two-dimensional representation of mutational signatures shown in S10 Fig, this plot indicates which patients had hypermethylation events in non-BRCA cancer-predisposition genes. Diamond shapes indicate patients for whom no loss-of-heterozygosity was observed.
(PDF)

**S35 Fig. Gene-expression levels for all the genes we studied.** For each gene, we identified tumors that expressed these genes at relatively low levels compared to other breast tumors; these low expressors are highlighted as red points.
(PDF)

**S36 Fig. Relationship between BRCA aberration status and relatively low gene expression.** We identified tumors with low expression for cancer-predisposition genes (see S35 Fig) and evaluated whether the somatic-mutation signatures of these tumors were relatively similar or dissimilar to the BRCA reference group. Each red rectangle represents a patient sample that expressed a given gene at low levels. Low expression of RAD51C and BRCA1 showed the strongest positive correlation between gene-expression status and the BRCAness reference group. Low expression of BARD1 and CDH1 showed the strongest negative correlation between gene-expression status and the BRCAness reference group. Genes for which no tumors exhibited low expression are omitted.
(PDF)

**S37 Fig. Relationship between BRCA aberration status and demographic, histopathological, and surgical observations in breast-cancer patients.** Red rectangles indicate patients that were positive for each respective clinical characteristic. Tumors with triple-negative hormone receptors, infiltrating ductal carcinoma histologies, or close surgical margins overlapped most with BRCA-aberrant tumors based on somatic-mutation signatures.
(PDF)

**S38 Fig. Relationship between BRCA aberration status and pharmacological responses in breast-cancer patients.** We evaluated clinical treatment responses for 211 TCGA patients for whom drug-response data were available. Responses for none of the drugs were significantly

correlated with BRCA aberration status based on somatic-mutation signatures.
(PDF)

**S1 Table. Outcome of pathogenicity evaluation for somatic mutations in BRCA1 and BRCA2.** We evaluated somatic mutations in BRCA1 and BRCA2 based on variant type, predicted effects on protein sequence, evolutionary conservation, minor allele frequency, evidence in ClinVar, etc. This table provides information about each variant and specified criteria that we considered. A value of 1 in the Pathogenicity column indicates that we considered the variant to be pathogenic in our analyses.
(DOCX)

**S2 Table. Summary of classification analysis for predicting a tumor's aberration status.** Via cross validation, we used gene-expression profiles and somatic-mutation signatures, respectively, to predict whether a given patient/tumor harbored particular types of aberrations. Sensitivity is equivalent to the true-positive rate. Specificity is equivalent to the true-negative rate. The area under the receiver operator characteristic curve (AUROC) quantifies the balance between sensitivity and specificity across a range of prediction thresholds.
(DOCX)

## Acknowledgments

Results from this study are in part based upon data generated by TCGA and managed by the United States National Cancer Institute and National Human Genome Research Institute (see http://cancergenome.nih.gov). We thank the patients who participated in this study and shared their data publicly. We thank the Fulton Supercomputing Laboratory at Brigham Young University for providing computational facilities.

## Author Contributions

**Conceptualization:** Weston R. Bodily, Brian H. Shirts, Stephen R. Piccolo.

**Data curation:** Alyssa Parker, Moom Roosan.

**Formal analysis:** Weston R. Bodily, Stephen R. Piccolo.

**Investigation:** Weston R. Bodily, Brian H. Shirts, Tom Walsh, Suleyman Gulsuner, Mary-Claire King, Stephen R. Piccolo.

**Methodology:** Stephen R. Piccolo.

**Supervision:** Stephen R. Piccolo.

**Visualization:** Stephen R. Piccolo.

**Writing – original draft:** Weston R. Bodily, Brian H. Shirts, Stephen R. Piccolo.

**Writing – review & editing:** Weston R. Bodily, Brian H. Shirts, Tom Walsh, Suleyman Gulsuner, Mary-Claire King, Alyssa Parker, Moom Roosan, Stephen R. Piccolo.

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
