## [Decision Letter · Decision Letter 0]

24 Jul 2020

PONE-D-20-16275

Effects of germline and somatic events in candidate BRCAness genes on breast-tumor signatures

PLOS ONE

Dear Dr. Piccolo,

Thank you for submitting your manuscript to PLOS ONE. After careful consideration, we feel that it has merit but does not fully meet PLOS ONE’s publication criteria as it currently stands. Therefore, we invite you to submit a revised version of the manuscript that addresses the points raised during the review process.

We look forward to receiving your revised manuscript.

Kind regards,

Alvaro Galli

Academic Editor

PLOS ONE

Journal Requirements:

"TW consults for Color Genomics. Otherwise, the authors declare that they have no competing interests."

Please respond by return email with your amended Competing Interests Statement and we will change the online submission form on your behalf.

4. Your ethics statement must appear in the Methods section of your manuscript. If your ethics statement is written in any section besides the Methods, please move it to the Methods section and delete it from any other section. Please also ensure that your ethics statement is included in your manuscript, as the ethics section of your online submission will not be published alongside your manuscript.

6. We note that currently the supporting figures in your supporting information file "BRCAness_Supplementary.docx" are not displaying. Can you please ensure all the supporting figures are included and display clearly?

Reviewers' comments:

Reviewer's Responses to Questions

**Comments to the Author**

1. Is the manuscript technically sound, and do the data support the conclusions?

Reviewer #1: Yes

Reviewer #2: Yes

2. Has the statistical analysis been performed appropriately and rigorously? 

Reviewer #1: Yes

Reviewer #2: Yes

3. Have the authors made all data underlying the findings in their manuscript fully available?

Reviewer #1: No

Reviewer #2: Yes

4. Is the manuscript presented in an intelligible fashion and written in standard English?

Reviewer #1: Yes

Reviewer #2: Yes

5. Review Comments to the Author

Reviewer #1: The authors have proved their findings by a bioinformatical and statistical analysis of several genetic samples obtained by public accessible databases. The statistical methods used seem appropriate for their conclusion and the statistical power coming from the samples considerated seems enough to prove their point.

1) My concern is that I was totally unable to visualize the supplementary figures and therefore I was unable to appropriately review the data discussed. I do not know if it was just my problem, but I would like to considerate also the supplementary figure before accepting the paper for publish.

2) In addition, I think that the method section needs more details and less references to other paper in order to allow the reader to reproduce the experiments and the analysis performed.

3) It would be interesting to see if applying their method to other candidate BRCAness genes such as the ones discovered by Konstantinopoulos et al (PMID 20547991), they would also fit into the group generated in this paper. Of course, if enough data are accessible from genetic databases.

Reviewer #2: In the present paper Bodily et al. present an analysis of molecular profiles of breast cancer data retrieved from TCGA in order to identify genes that can be associated with BRCAness. Firstly they analyzed data of patients carrying BRCA1/2 germline mutation to define the molecular features in term of somatic mutational signature and expression profile. Then they analyzed molecular profiles of breast cancer patients carrying somatic BRCA1/2 alteration (promoter methylation, somatic mutation, homozygous deletion). They observed that BRCA1/2 germiline and somatic carriers have homogenous mutational signature and expression profile so they use all group as reference group to evaluate whether molecular aberrations in cancer–predisposing genes determine mutational signatures similar to breast cancer carrying BRCA1/2 alterations.

Genomic approaches are very important to address fundamental biological questions of breast cancer and in particular, new studies to better define BRCAness are necessary for the identification of new therapeutic approaches. The approach used in the manuscript is intriguing and has the potential to positively influence the field of BRCAness. In my opinion however, the manuscript contains critical pitfalls that renders it unsuitable for publication in the present form. I listed below comments.

Major comments

1) Supplementary data: The figures of supplementary file could not be opened with Word of Office (I tried with different version). I was able to open them with LibreOffice. Unfortunately supplementary tables were not even seen with LibreOffice.

2) Line 40: “suggests additional genes”. What genes? ATR and BARD1? Are really new? Too vague sentence. In the paper of Lord and Asworth 2016 (BRCAness revisted) ATR is already included among BRCAness genes.

3) Section methods “data preparation and filtering” should be divided into paragraphs to distinguish how the authors analyzed the several molecular features.

4) Line 120-121: The description of selection parameter for somatic mutation is unclear. Instead of “we used following criteria to exclude somatic variant” I would say somatic variants that are 1)… 2) were excluded

5) Also for results section the author did not divided into paragraphs. In this way different results are difficult to follow and the findings relative to each approach is lost during the reading. In the first part of results (line 204-215) the authors have written down an outline of the analysis they have done but on the followings paragraphs there is a mix. For instance homogeneity is treated at line 239-244 and then at line 251-258. Since the conclusion is that BRCA1 and BRCA2 germline carriers show homogenous somatic mutation signature and expression profile, it would result clear to join the two paragraph and add a title. Paragraphs with titles would help the reader to understand the results.

The fragmentation of results is also found in the figures. In line 217 is described Figure1A while the other panels of figure 1 are described in line 265. In my opinion, is easier to understand the paper if the panels of the figure are described in the same paragraph.

I would group the results in this way adding the following paragraphs (titles are just an example):

Aberration in BRCA1 and BRCA2

Line 216-225

Line 245-250

Line 264-277

Expression profile and signature of breast cancer

lines 226-238: It is not clearly specified if this analysis is for all breast cancer (1101 patients) or for BRCA1/2 germline carries. It seems that this analysis refers to all patients. Why is it described after the analysis of germlines BRCA1/2? May be it is better as first paragraph.

Homogeneity of somatic mutation signature and expression profile of germline BRCA1/2 carriers

lines 239-244; 251-258.

Similarity between BRCA1/2 germline carriers

lines 259-262

Aberration in cancer predisposing genes

Lines 283-303

Line 305-318

6) Discussion: The discussion section overall lacks references and in some parts is a mere description of results. For instance how can be explained affirmation of lines 327-331? What has it been shown in previous papers? What is the impact of result of paper on BRCAness definition? Could the result help in finding new therapeutic approaches?

7) Line 369: which are the new factors highlighted? Explain better. What do you mean for factors? The two genes identified? Are they really new?

Minor comments:

1) Line 31 After “somatic-mutation signatures of tumors having” I would add molecular aberrations such as ….

2) Lane 170 extreme values instead of extremevalues

3) Line 293 for 11 genes …add the list of genes to guide the reader in the analysis of the table

6. PLOS authors have the option to publish the peer review history of their article (what does this mean?). If published, this will include your full peer review and any attached files.

Reviewer #1: No

Reviewer #2: No

---

## [Author Response · Author response to Decision Letter 0]

13 Aug 2020

Dear editors:

Thank you for reviewing our manuscript, "Markers of BRCAness in breast cancer." We apologize for the technical difficulties with the supplementary section of our manuscript. We have provided the supplementary tables and figures as separate files to avoid further complications. Below we provide a detailed response to the editor's and reviewers' comments.

> We have addressed these requirements. We are happy to address anything that we may have missed.

"TW consults for Color Genomics. Otherwise, the authors declare that they have no competing interests."

> We have added the phrase, "This does not alter our adherence to PLOS ONE policies on sharing data and materials," to our Competing Interests section.

Please respond by return email with your amended Competing Interests Statement and we will change the online submission form on your behalf.

> We have responded by email with our amended Competing Interests statement.

> Our analysis uses germline-variant data from The Cancer Genome Atlas (TCGA) human cohort. The TCGA Human Subjects Protection and Data Access Policies state the following: "The controlled-access data tier will not be freely available to the public, but will be made available to any qualified researcher for the purpose of biomedical research, once the investigator, along with his/her institution, has certified agreement to the statements within TCGA Data Use Certification (DUC). The data types in the controlled access tier include...individual-level germline variant data." (https://www.cancer.gov/about-nci/organization/ccg/research/structural-genomics/tcga/history/policies/tcga-human-subjects-data-policies.pdf) In addition, this page provides information about how researchers may contact the TCGA data-access committee with any questions. Accordingly, researchers can obtain germline variant data for the TCGA breast-cancer cohort, as we did. To make re-analysis more convenient for other researchers, we would be happy to share our summarized, filtered variant calls with any researcher who requests them, as long as the researcher has obtained approval to access TCGA controlled data. We have not and will not place any restrictions on data access beyond those of the TCGA data-access committee. Such restrictions are common for germline data. Accordingly, we believe we are in compliance with PLOS policies regarding data sharing.

> Thank you for updating this statement on our behalf. Please let me know if I can provide any other useful information regarding data availability. 

4. Your ethics statement must appear in the Methods section of your manuscript. If your ethics statement is written in any section besides the Methods, please move it to the Methods section and delete it from any other section. Please also ensure that your ethics statement is included in your manuscript, as the ethics section of your online submission will not be published alongside your manuscript.

> We have moved our ethics statement to the Methods section.

> We have moved the captions for the Supporting Information files to the end of our manuscript and updated the in-text citations to match them.

6. We note that currently the supporting figures in your supporting information file "BRCAness_Supplementary.docx" are not displaying. Can you please ensure all the supporting figures are included and display clearly?

> Thank you, and we apologize. We have fixed this problem and now provide these figures and tables as separate files to avoid further complications.

Reviewers' comments:

Reviewer's Responses to Questions

Comments to the Author

1. Is the manuscript technically sound, and do the data support the conclusions?

Reviewer #1: Yes

Reviewer #2: Yes

> Thank you.

2. Has the statistical analysis been performed appropriately and rigorously?

Reviewer #1: Yes

Reviewer #2: Yes

> Thank you.

3. Have the authors made all data underlying the findings in their manuscript fully available?

Reviewer #1: No

Reviewer #2: Yes

> Our analysis uses germline-variant data from The Cancer Genome Atlas (TCGA) human cohort. The TCGA Human Subjects Protection and Data Access Policies state the following: "The controlled-access data tier will not be freely available to the public, but will be made available to any qualified researcher for the purpose of biomedical research, once the investigator, along with his/her institution, has certified agreement to the statements within TCGA Data Use Certification (DUC). The data types in the controlled access tier include...individual-level germline variant data." (https://www.cancer.gov/about-nci/organization/ccg/research/structural-genomics/tcga/history/policies/tcga-human-subjects-data-policies.pdf) In addition, this page provides information about how researchers may contact the TCGA data-access committee with any questions. Accordingly, researchers can obtain germline variant data for the TCGA breast-cancer cohort, as we did. To make re-analysis more convenient for other researchers, we would be happy to share our summarized, filtered variant calls with any researcher who requests them, as long as the researcher has obtained approval to access TCGA controlled data (in general). We have not and will not place any restrictions on data access beyond those of the TCGA data-access committee. Such restrictions are common for germline data. Accordingly, we believe we are in compliance with PLOS policies regarding data sharing.

4. Is the manuscript presented in an intelligible fashion and written in standard English?

Reviewer #1: Yes

Reviewer #2: Yes

> Thank you. 

5. Review Comments to the Author

Reviewer #1: The authors have proved their findings by a bioinformatical and statistical analysis of several genetic samples obtained by public accessible databases. The statistical methods used seem appropriate for their conclusion and the statistical power coming from the samples considerated seems enough to prove their point.

> We thank the reviewer for taking the time to review the article and for providing a timely response!

1) My concern is that I was totally unable to visualize the supplementary figures and therefore I was unable to appropriately review the data discussed. I do not know if it was just my problem, but I would like to considerate also the supplementary figure before accepting the paper for publish.

> It was a technical glitch that we, as the authors, should have prevented. We apologize. In the current version of the manuscript, we have uploaded these figures and tables as separate files to avoid such problems.

2) In addition, I think that the method section needs more details and less references to other paper in order to allow the reader to reproduce the experiments and the analysis performed.

> Thank you for this suggestion. We have added methodological details to various paragraphs throughout the Methods section. In place of prior sentences that referred the reader to referenced papers for these details, we have provided more details so that the reader can understand our approach without necessarily needing to read those other papers. Additionally, as before, we have provided a GitHub repository and an Open Science Framework repository that provide code that we used as an additional reference in support of reproducibility.

3) It would be interesting to see if applying their method to other candidate BRCAness genes such as the ones discovered by Konstantinopoulos et al (PMID 20547991), they would also fit into the group generated in this paper. Of course, if enough data are accessible from genetic databases.

> Konstantinopoulos, et al. identified a signature of 60 genes that they used to classify patients into BRCA-like and non-BRCA-like categories based on expression levels of those genes. The reviewer suggested that these genes might also be considered as candidate BRCAness genes. However, to do this properly, we would need to have germline mutation status for these genes, but our process for determining pathogenicity of germline mutations is limited to the candidate genes that we evaluated in the paper, so I'm afraid it would not be a comprehensive comparison. Furthermore, the findings of Konstantinopoulos, et al. do not necessarily suggest that DNA mutations, copy-number variations, or hypermethylation of these genes are drivers of BRCAness; rather their findings suggest that these genes can be used to categorize patients into the BRCAness category, irrespective of the underlying driver events. Konstantinopoulos, et al. showed that their expression signature was successful at predicting response to a platinum agent and a PARP-inhibitor in samples relevant to epithelial ovarian cancer. We attempted to test this signature using the limited drug-sensitivity data available for the breast-cancer patients in TCGA. However, we had drug-sensitivity data for a single platinum agent (Carboplatin) and for no PARP inhibitors. For Carboplatin, we had data for 9 patients but only 2 non-responders, which was an insufficient sample size for building a predictive model. We hope that in the future, it will be possible to do more analysis with drug sensitivity data.

Reviewer #2: In the present paper Bodily et al. present an analysis of molecular profiles of breast cancer data retrieved from TCGA in order to identify genes that can be associated with BRCAness. Firstly they analyzed data of patients carrying BRCA1/2 germline mutation to define the molecular features in term of somatic mutational signature and expression profile. Then they analyzed molecular profiles of breast cancer patients carrying somatic BRCA1/2 alteration (promoter methylation, somatic mutation, homozygous deletion). They observed that BRCA1/2 germiline and somatic carriers have homogenous mutational signature and expression profile so they use all group as reference group to evaluate whether molecular aberrations in cancer–predisposing genes determine mutational signatures similar to breast cancer carrying BRCA1/2 alterations.

Genomic approaches are very important to address fundamental biological questions of breast cancer and in particular, new studies to better define BRCAness are necessary for the identification of new therapeutic approaches. The approach used in the manuscript is intriguing and has the potential to positively influence the field of BRCAness. In my opinion however, the manuscript contains critical pitfalls that renders it unsuitable for publication in the present form. I listed below comments.

> We thank the reviewer for taking the time to review the article and for providing a timely response!

Major comments

1) Supplementary data: The figures of supplementary file could not be opened with Word of Office (I tried with different version). I was able to open them with LibreOffice. Unfortunately supplementary tables were not even seen with LibreOffice.

> It was a technical glitch that we, as the authors, should have prevented. We apologize. In the current version of the manuscript, we have uploaded these figures and tables as separate files to avoid such problems.

2) Line 40: “suggests additional genes”. What genes? ATR and BARD1? Are really new? Too vague sentence. In the paper of Lord and Asworth 2016 (BRCAness revisted) ATR is already included among BRCAness genes.

> The reviewer makes an important point. Earlier in the abstract, we mention ATR and BARD1 as well as other genes that "showed high similarity but only for a small number of events or for a single event type." We have clarified this part to specifically mention the genes to which we were referring. Having clarified this part, we revised the final sentence so that instead of reiterating which genes were significant and suggesting that they might be considered for inclusion in the BRCAness definition (some of which already have been, as the reviewer notes), we emphasize that our "methodology represents an objective way to identify genes that have similar downstream effects on molecular signatures when mutated, deleted, or hypermethylated."

3) Section methods “data preparation and filtering” should be divided into paragraphs to distinguish how the authors analyzed the several molecular features.

> We have divided this section into additional paragraphs to make the section more readable.

4) Line 120-121: The description of selection parameter for somatic mutation is unclear. Instead of “we used following criteria to exclude somatic variant” I would say somatic variants that are 1)… 2) were excluded

> Thank you. We have made this change.

5) Also for results section the author did not divided into paragraphs. In this way different results are difficult to follow and the findings relative to each approach is lost during the reading. In the first part of results (line 204-215) the authors have written down an outline of the analysis they have done but on the followings paragraphs there is a mix. For instance homogeneity is treated at line 239-244 and then at line 251-258. Since the conclusion is that BRCA1 and BRCA2 germline carriers show homogenous somatic mutation signature and expression profile, it would result clear to join the two paragraph and add a title. Paragraphs with titles would help the reader to understand the results.

> Thank you for these helpful observations. We have reorganized the text and added two figures to help convey the results in a more logical and organized manner. Below we provide additional responses to these suggestions.

The fragmentation of results is also found in the figures. In line 217 is described Figure1A while the other panels of figure 1 are described in line 265. In my opinion, is easier to understand the paper if the panels of the figure are described in the same paragraph.

> After reorganizing the Results section, the references to Figure 1 are now in the same paragraph. 

lines 226-238: It is not clearly specified if this analysis is for all breast cancer (1101 patients) or for BRCA1/2 germline carries. It seems that this analysis refers to all patients. Why is it described after the analysis of germlines BRCA1/2? May be it is better as first paragraph.

> We have clarified that this applies to all breast-cancer patients in the cohort and have added two figures that illustrate these data across all patients.

I would group the results in this way adding the following paragraphs (titles are just an example):

Aberration in BRCA1 and BRCA2

Line 216-225

Line 245-250

Line 264-277

Expression profile and signature of breast cancer

lines 226-238

Homogeneity of somatic mutation signature and expression profile of germline BRCA1/2 carriers

lines 239-244; 251-258.

Similarity between BRCA1/2 germline carriers

lines 259-262

Aberration in cancer predisposing genes

Lines 283-303

Line 305-318

> We followed the reviewer's advice, although we have used somewhat different section names and have structured the subsections slightly differently than what the reviewer recommended. But we feel that our changes address the spirit of what the reviewer recommended.

6) Discussion: The discussion section overall lacks references and in some parts is a mere description of results. For instance how can be explained affirmation of lines 327-331? What has it been shown in previous papers? What is the impact of result of paper on BRCAness definition? Could the result help in finding new therapeutic approaches?

> Thank you for these suggestions. We have rewritten the Discussion section substantially in response to the reviewer's comments and questions. We have removed parts that were mere descriptions of results, and we have added citations in this section. We now provide some commentary on the utility of our approach for helping to clarify the definition of BRCAness and potentially to better understand mechanisms of BRCAness.

7) Line 369: which are the new factors highlighted? Explain better. What do you mean for factors? The two genes identified? Are they really new?

> Thank you for pointing this out. Our wording was too vague. As mentioned above, we have rewritten the Discussion section to provide more thorough descriptions of prior evidence associated with our results. As part of this, we are now more explicit about the literature surrounding BARD1 and ATR, in particular.

Minor comments:

1) Line 31 After “somatic-mutation signatures of tumors having” I would add molecular aberrations such as ….

> Thank you. We have made this change.

2) Lane 170 extreme values instead of extremevalues

> The name of the package is "extremevalues" without any spaces.

3) Line 293 for 11 genes …add the list of genes to guide the reader in the analysis of the table

> Thank you. We have made this change.

6. PLOS authors have the option to publish the peer review history of their article (what does this mean?). If published, this will include your full peer review and any attached files. Do you want your identity to be public for this peer review? For information about this choice, including consent withdrawal, please see our Privacy Policy. 

Reviewer #1: No

Reviewer #2: No

> Thank you.

> We have performed this check. PACE reported no issues. In our resubmission, we used the TIFF files created by PACE for the non-supplementary figures.

---

## [Decision Letter · Decision Letter 1]

2 Sep 2020

Effects of germline and somatic events in candidate BRCAness genes on breast-tumor signatures

PONE-D-20-16275R1

Dear Dr. Piccolo,

We’re pleased to inform you that your manuscript has been judged scientifically suitable for publication and will be formally accepted for publication once it meets all outstanding technical requirements.

Kind regards,

Alvaro Galli

Academic Editor

PLOS ONE

Additional Editor Comments (optional):

Reviewers' comments:

Reviewer's Responses to Questions

**Comments to the Author**

1. If the authors have adequately addressed your comments raised in a previous round of review and you feel that this manuscript is now acceptable for publication, you may indicate that here to bypass the “Comments to the Author” section, enter your conflict of interest statement in the “Confidential to Editor” section, and submit your "Accept" recommendation.

Reviewer #1: All comments have been addressed

Reviewer #2: All comments have been addressed

2. Is the manuscript technically sound, and do the data support the conclusions?

Reviewer #1: Yes

Reviewer #2: Yes

3. Has the statistical analysis been performed appropriately and rigorously? 

Reviewer #1: Yes

Reviewer #2: Yes

4. Have the authors made all data underlying the findings in their manuscript fully available?

Reviewer #1: Yes

Reviewer #2: Yes

5. Is the manuscript presented in an intelligible fashion and written in standard English?

Reviewer #1: Yes

Reviewer #2: Yes

6. Review Comments to the Author

Reviewer #1: The authors have addressed my concerns and I thereby suggest this paper for publication.

I would like to thank the authors to answering my questions and modifying the paper with my suggestions.

Reviewer #2: Dear Editor,

the revised version of the paper is greatly improved. The new organization of the paper and the exhaustive discussion make it clearer and interesting. The authors have done a fine job of responding to reviewer’s comments.

I noticed a typo at line 490 the word “genmic” is used instead of genomic.

7. PLOS authors have the option to publish the peer review history of their article (what does this mean?). If published, this will include your full peer review and any attached files.

Reviewer #1: No

Reviewer #2: No

---

## [Editor Report · Acceptance letter]

9 Sep 2020

PONE-D-20-16275R1 

Effects of germline and somatic events in candidate BRCA-like genes on breast-tumor signatures 

Dear Dr. Piccolo:

I'm pleased to inform you that your manuscript has been deemed suitable for publication in PLOS ONE. Congratulations! Your manuscript is now with our production department. 

Kind regards, 

on behalf of

Dr. Alvaro Galli 

Academic Editor

PLOS ONE